# Discussing surgical innovation with patients: a qualitative study of surgeons' and governance representatives' views

Jesmond Zahra [1,2] Sangeetha Paramasivan,[1,2] Natalie S Blencowe [1,2,3] Sian Cousins,[1,2] Kerry Avery [1,2] Johnny Mathews,[1,2] Barry G Main,[1,2,4] Angus G K McNair [1,2,5] Robert Hinchliffe,[1,2,6] Jane M Blazeby,[1,2,4] Daisy Elliott[1,2]

For numbered affiliations see end of article.

**Correspondence to**
Dr Daisy Elliott;
daisy.elliott@bristol.ac.uk

## ABSTRACT

**Objectives** Little is known about how innovative surgical procedures are introduced and discussed with patients. This qualitative study aimed to explore perspectives on information provision and consent prior to innovative surgical procedures.

**Design** Qualitative study involving semi-structured interviews. Interviews were audio recorded, transcribed and analysed thematically.

**Participants** 42 interviews were conducted (26 surgeons and 16 governance representatives).

**Setting** Surgeons and governance representatives recruited from various surgical specialties and National Health Service (NHS) Trusts across England, UK.

**Results** Participants stated that if a procedure was innovative, patients should be provided with additional information extending beyond that given during routine surgical consultations. However, difficulty defining innovation had implications for whether patients were informed about novel components of surgery and how the procedure was introduced (ie, as part of a research study, trust approval or in routine clinical practice). Furthermore, data suggest surgeons found it difficult to establish what information is essential and how much detail is sufficient, and governance surrounding written and verbal information provision differed between NHS Trusts. Generally, surgeons believed patients held a view that 'new' was best and reported that managing these expectations could be difficult, particularly if patient views aligned with their own.

**Conclusions** This study highlights the challenges of information provision and obtaining informed consent in the context of innovative surgery, including establishing if and how a procedure is truly innovative, determining the key information to discuss with patients, ensuring information provision is objective and balanced, and managing patient expectations and preferences. This suggests that surgeons may require support and training to discuss novel procedures with patients. Further work should capture consultations where new procedures are discussed with patients and patients' views of these information exchanges.

## Strengths and limitations of this study

► This is the first qualitative study to conduct in-depth, semi-structured interviews to understand current practice for information provision and informed consent for innovative surgical procedures in the UK.
► Interviews with governance representatives enabled us to understand what policies were in place for the introduction of new procedures, while surgeons provided insights into how they introduced new procedures and discussed these with patients.
► Purposeful sampling ensured that participants were recruited from a range of geographical locations and trust types, and surgeons were from different specialties with varying experiences of innovation.
► Audio-recording appointments where clinicians discuss innovative surgeries and patients decide whether or not to undergo the procedure would strengthen our understanding of interactions between clinicians and patients.
► Further work should capture patients' views on information provision in this context, so that recommendations can be made to improve transparency and communication.

modern healthcare provision. Until recently, the Bolam ruling meant that a doctor's duty to inform patients prior to surgery was judged based on whether they had acted in line with the view of a responsible body of medical opinion.[1 2] The 2015 Montgomery ruling[3] redefined the standard of informed consent in the UK and represented a shift from a paternalistic model of consent to a more patient-centred approach[4 5]—meaning that a patient should be told whatever they want to know, not what the doctor thinks they should be told.[5]

Recently, important questions have been raised surrounding informed consent for innovative surgical procedures.[6 7] An inquest into the death of the first patient in the UK to undergo robotically assisted heart surgery found the patient had not been fully informed

## INTRODUCTION

Informed consent is fundamental to patient autonomy and represents a cornerstone of

about the comparative risk of robotic versus conventional open surgery.[7] In addition, a summary of invited reviews published by the Royal College of Surgeons of England (RCSEng) raised further concerns.[6] The review highlighted issues surrounding surgeons' ability to accurately quantify risk as reported in published literature, whether surgical experience is being sufficiently disclosed and effectively communicated to patients, and the overall quality of patient–clinician discussions.[6] The RESEng subsequently published guidelines for the development, implementation and dissemination of surgical innovation, which contain guidance on informed consent.[8] This suggests, based on findings of a review by Broekman *et al*,[8] that patients should be informed of the innovative nature of the procedure, surgeons' experience with the procedure and the learning curve, the risk and benefits—including unknown or unforeseeable risks or outcomes, the evidence or lack thereof, and alternatives to the innovative procedure.

Reviews and commentaries have highlighted potential issues around informed consent for innovative surgical procedures, including patients' and surgeons' beliefs that new treatments constitute improved treatment options or result in better outcomes.[9 10] However, little is known about current practice. Surveys that have been conducted in the USA suggest that information varies considerably in content and quality. Reitsma and Moreno reported that of 21 surgeons, 75% verbally informed patients about the innovative nature of the procedure but only 33% provided written information.[11] Lee *et al*, in a scenario-based survey carried out with 85 surgeons and 383 patients, found differences between what surgeons thought patients wanted to know about innovative procedures and what patients reported they wanted to know. For instance, compared with surgeons, patients placed more importance on nearly all types of information, particularly volumes and outcomes.[12]

While a handful of qualitative studies have been conducted exploring how innovative procedures have been introduced,[13–16] none have specifically looked at information provision and informed consent. Moreover, these studies were conducted in Australia and Canada and findings may not be generalisable to other countries and different healthcare systems.[17] New invasive procedures may be introduced in the context of formal research studies or via local hospital policies.[18] The National Institute for Health and Care Excellence (NICE) recommends that local National Health Service (NHS) organisations have appropriate governance structures in place to review, approve and monitor the introduction of new invasive procedures.[19] The current study sought to explore surgeons' and governance representatives' views of information provision and informed consent for the introduction of new invasive procedures in the UK.

## METHODS

### Patient and public involvement

Public and patient involvement was not conducted as part of the current study. However, the current findings have been discussed with a recently established group, which has led to further work which specifically aims to explore patient views on surgical innovation.

### Design

This qualitative study employed a grounded theory methodology[20] because it enabled the inductive identification of codes from the data to generate new hypotheses about phenomena that are derived or grounded in the data.[21] Its central principle is of constant comparison, where new findings are systematically compared with existing data so that similarities and differences can be identified and emerging theories refined through the ongoing assimilation of data.[20 22] Consistent with grounded theory methodology, semi-structured interviews were conducted with relevant stakeholders.

Ethical approval was granted by the University of Bristol Faculty of Health Sciences Research Ethics Committee. The study is reported in line with the Standards for Reporting Qualitative Research (online supplemental file 1).[23]

### Sampling and recruitment

Participants were recruited between November 2017 and October 2018 using a combination of purposive[24] and snowball sampling techniques.[25] Governance representatives (defined as those involved in regulating the introduction of new/modified procedures and/or devices at trust or national levels) were included to understand what processes were in place for the introduction of new procedures.[16] Surgeons were recruited to explore experiences of introducing new/modified procedures or devices into clinical practice. As there is no standardised definition for innovation and it is often a continuum from standard practice through to true innovation,[13 26 27] and because surgical innovation is unlikely to involve a single discrete development,[28 29] participants were asked to reflect what the term meant to them and describe procedures that they deemed to be innovative.

Governance representatives were identified through websites (eg, NHS Trusts websites) and NHS policy documents (eg, new invasive procedure/devices policies). An initial sample of surgeons was identified by academic surgeons working within the Centre for Surgical Research, University of Bristol, UK. Subsequent surgeons were identified via snowball sampling,[25] in which interviewees recommended others who may be willing to take part. Sampling was regularly reviewed to ensure surgeons from different specialties and geographical locations, with varying experiences of innovation (eg, minor modifications vs first in human procedures), were recruited to capture a diversity of perspectives.[30 31] For governance representatives, maximum variation was sought in relation to different roles (eg, new procedures committee

members, medical directors), geographical locations, trust types and different surgical specialties.

Potential participants were approached via email and up to three invitation emails were sent before contact attempts ceased. In total, 83 potential participants were invited to take part (38 surgeons and 45 governance representatives). Study recruitment ceased when the qualitative team agreed that theoretical saturation had been achieved (where no new themes were forthcoming from three consecutive interviews).[32 33]

## Data collection

Interviews were considered to be the most appropriate method for this study because they provided the opportunity to encourage participants to think carefully about their own experiences[34] and enable the interviewer to respond and follow up on issues raised by the interviewee.[35] Interviews were chosen instead of focus groups as it was felt that some individuals may feel intimidated at the prospect of discussing their experiences within a focus group setting.[25]

All participants received a study information leaflet and provided written consent before taking part. Separate topic guides were developed for governance representatives and surgeons. All interview schedules contained core topics of interest (eg, 'How would you define innovation?', 'What does informed consent mean to you?'),[36 37] although these were flexible to allow participants to shape the direction of conversations.[38 39] Governance representatives were asked how applications for new invasive procedures were processed and how these procedures then integrated into practice, whereas surgeons were asked about their experiences of introducing procedures and what patients had been told. An example of each interview schedule is provided in the online supplemental files 2 and 3.

The interviewer began the discussion with more general questions to build rapport[40] such as the participants' background and general views of defining innovation. The interviewee's comments were restated and incorporated into further questions to check with the interviewer that they had understood correctly.[38] In order to gain more detailed and comprehensive understanding of the descriptions given by participants, they were asked to elaborate their comments with explanations and examples (such as, 'Why do you feel this was?'),[38] or fillers (such as 'mmm' or 'yeah') were used by the interviewer to show the participant this was something they were interested in and to encourage them to continue.[41] Towards the end of the interview, participants were given an opportunity to raise issues that were important to them that had not already been covered.[40]

Topic guides were developed iteratively during the study period, in consideration of emerging insights from interviews and better ways of phrasing questions.[36 42] For instance, interviewers (non-clinical academics) used more prompts to encourage the surgeons to describe procedures in a non-technical manner.[38] Topic guides were initially very long so were reduced to include more 'mapping' questions that addressed the core components of interest (such as, 'I understand you've conceptualised a new procedure. Can you talk me through this?'). Additional prompts were added from emerging findings (ie, colleagues' reactions to procedures and written information). With permission, interviews were audio recorded using an encrypted digital dictaphone.

All interviews were audio recorded except one where the interviewee requested not to be recorded and therefore written notes were taken. All interviews were conducted by one of two trained and experienced qualitative researchers (JZ or DE). JZ is a male senior research associate with a background in public health research. DE is a female research fellow with experience working on surgical trials. She is also a member of the QuinteT research group, which uses qualitative research methods to optimise recruitment and informed consent to randomised controlled trials (RCTs). JZ and DE have several years of experience conducting qualitative research, and each has a PhD in health-related research.

## Data analysis

Interview recordings were transcribed and de-identified so that names and identifiable information were removed. Data were imported into a data management program (NVivo V.11) to organise the data systematically, allow for quick searching and refining of codes and categories, increase the transparency of the analysis process and enable team comparisons of coding.[32 43] Thematic analysis was undertaken using the constant comparison technique of grounded theory.[20 44] First, analysts (JZ, DE and SP) read and reread transcripts before independently coding a sample of five transcripts to begin familiarising themselves with the data.[32 45] Interesting features were coded, whereby a segment of data was assigned a code,[42] and ongoing potential ideas were noted at every stage of analysis.[46] Data coding was compared to ensure consistency and rigour in the findings,[47] whereby differences were discussed thoroughly until a consensus was met and an initial coding frame was developed. The coding frame was then independently applied to three transcripts by JZ, DE and SP. High consistency between analysts indicated the coding frame appropriately captured the data. JZ then applied the coding frame to the remaining transcripts, adding additional codes if important data were missed by the original coding frame. To form preliminary themes and subthemes, codes were collated into hierarchical groups.[48] Emerging findings were regularly discussed with JMB and NSB (both academic surgeons), with reference to the raw transcripts, to ensure they fully encapsulated the meaning of the data.[47] Finally, a descriptive account of the themes and subthemes, which included representative quotes, was developed.

## RESULTS

### Participants

Of the 83 participants that were approached, 42 participants were recruited (those who did not participate were unable to find time for an interview or did not respond to study invites). The final sample included 26 surgeons and 16 governance representatives from across 23 NHS Trusts. Trusts varied in geographical location, type/size (ie, teaching, specialist, large, multicentre) and foundation status. Governance representatives' roles within trusts and NICE varied (eg, medical director, director for quality improvement, head of governance), however, all played a role in regulating the introduction of new surgical procedures and/or devices. At the time of interview, all but one (retired) surgeon worked as a consultant within the NHS. Surgical specialties included cardiothoracic, gastrointestinal, breast, ophthalmology, neurosurgery, urology and orthopaedics. Participants were mostly men (87%). Interviews were conducted over the phone (55%) or face-to-face at participants' place of work or at a university. Interviews lasted an average of 43 min (range 22–112 min, SD 17 min).

### Analysis

Findings related to informed consent are presented under four main themes: difficulty defining innovation, differing views on what—and how—patients should be told, the challenges of discussing innovation and managing patient expectations. All quotations are followed by participants' unique identification number, gender, surgical specialty and trust identification number.

### Difficulty defining innovation

Surgeons and governance representatives found it challenging to define surgical innovation and considered the term 'innovation' to encompass a spectrum of changes to surgical healthcare. Examples ranged from changing patient pathways through to first in human procedures. Consequently many felt innovation should be viewed as a continuum, rather than a binary concept:

> Well there's a spectrum isn't there, I mean it's difficult to define. So something that is innovative to me is something that is new, it's a new way of looking at it, it's a slightly different paradigm, a different way of thinking. And an adaptation is just a sort of a natural evolution of something that's already pre-existing, I suppose that's how I would define them, but clearly it's a spectrum isn't it? (Surgeon 18, General Surgery, Male, Trust 3)

There was variation in how the procedure had been introduced in participants' hospitals. Within this, surgeons' own experiences included requesting approval from their trust's new procedure committee, conducting the procedure in the context of a research study or deeming that approval was not necessary as the procedure was a variation of an existing one (from either judgement of the surgeon or confirmation from a committee). Table 1 highlights that the challenges of defining innovation, alongside the required technical knowledge of the procedure, could cause uncertainty as to what governance was required.

Although participants initially stated that patients should be informed about new procedures, some drew

**Table 1** Participants' experiences of governance for new procedures

| | Example quote |
|---|---|
| Procedure considered and approved by trust's new procedure committee | "There's one surgeon in [European country] who described the procedure, and the surgeon in [Asian country] who had done some of these procedures, but not really clear on how they'd done it. I contacted the [European] surgeon. He gave me some pointers, so I discussed it with my colleagues, and of course put it through the governance processes as a novel and a new procedure. And then we did it." (Surgeon 31, Colorectal, Male, Trust 8) |
| Approval from trust's new procedure committee deemed unnecessary as felt to be a variation of established procedure | "We asked about it to the new procedures committee, but they said well it's basically [variation of established procedure] isn't it. The technicalities of it are lost on non-thoracic surgeons really, which is good." (Surgeon 31, General Surgery, Male, Trust 8) "There's a new technique that's probably done by no more than 9 or 10 surgeons in the country. Again, it's a new technique, it's only been reported in the last 3 or 4 years. (…)Because it was under the umbrella of [broader procedure], I didn't have to go through any, so within the hospital there's a board about new techniques, you have to sort of, clinical effectiveness group, you have to have that approved. But, because this was a variation, we didn't need that approval." (Surgeon 22, Male, Orthopaedics, Trust 7) |
| Procedure evaluated in the context of a research study | "We were looking [procedure], and so that was in the guise of a randomised double-blind trial that we put through Ethics. So that would be innovative. We've got an Ethics Committee, and as part of that if you were going to be doing something really brand new that would have to be conducted as part of a trial, so, but when there's a blur between doing something slightly different then no I don't think it's the place for that(…)I can't think of anything where they've been involved in, you know, anything that we're doing." (Surgeon 46, Male, Orthopaedics, Trust 22) |

on experiences that contradicted this as they reflected on the complexity of defining innovation:

> I think absolutely they should be informed, I think that's a tricky one actually… when would I inform a patient about a new instrument I was using? So we don't- if- if you're not trialling the instrument to see if it's safe or more effective or less effective, then I don't think they need to be informed. (Surgeon 4, Male, General and Upper Gastrointestinal, Trust 12)

> We kind of created our own version of how you do that operation […] So, it's not particularly been described before by surgeons and it's not the world's most different, but it's got a few little innovative techniques. So, that was an operation we designed that was new […] I don't even explain how that's different to [standard procedure] because I think it's far too technically nuanced for a patient to understand… We view it as just a variation of [standard procedure]. (Surgeon 22, Thoracic, Male, Trust 7)

Several governance representatives also described instances where they had become aware of patients not being fully informed about new procedures and emphasised the need for improved information provision:

> So we then audit the first case to make sure that the clinician has been up front with the individual about the fact that this is the first time that this particular procedure has been undertaken…if they haven't, there's been a couple of occasions when they haven't, we tell them off and say do better next time and we audit the second case or whatever, so we're confident they've got it right. (Governance Representative 29, Male, Trust 5)

> We have had a couple of near misses … one was a surgeon inviting another surgeon from another hospital to come and show him how to do a procedure, that person did not have an honorary contract with the Trust therefore shouldn't have been operating and nobody knew that that was occurring. And we have had a rep [industry representative] coming into theatres to bring some samples of a new piece of equipment and most certainly patients weren't consented that it was going to be used on them … So I think there is increasing awareness now, certainly when business cases are presented, they do need to follow this process if that's what's going to happen. However there is also, can I put my hand on my heart and say actually the surgeon's not going to invite a rep into theatre tomorrow? I probably couldn't, but I do know that the theatre staff feel more empowered to be able challenge that behaviour, so actually if I knew that somebody was going to use a piece of equipment they've never used before I would probably be asking the question as to why. (Surgeon 9, Upper Gastrointestinal, Male, Trust 3)

### Differing views on what—and how—patients should be told

There was consensus from all participants that if a procedure was deemed innovative, patients should be provided with additional information extending beyond that given during routine surgical consultations. Within this, the potential risks and benefits, alternative treatment options and the novel status of the procedure were universally regarded as essential information that should be discussed.

> We tell them that it's a new procedure and previously we've told them how many we've done prior to this (Surgeon 5, Thoracic, Male, Trust 5)

> You have to give them options, you have to let them know what the evidence base is. (Surgeon 46, Trauma and Orthopaedic, Male, Trust 22)

There was variation in views as to whether other information—such as the evidence base (or lack thereof), operative experience, training, safety precautions implemented in theatre and national guidance—should be disclosed. Taken together, there appeared to be uncertainty at precisely how much to tell patients:

> Sometimes I tell them too much or tell them… I mean, the latest, er, ruling of Montgomery as I understand it is that they need to be told what they would want to know. Bloody hell. But then you've got to judge what they want to know and you've got to have some kind of communication with them and reasonably understand what it is they want to know. (Surgeon 31, Colorectal, Male, Trust 8)

> There is a difficult balance I think, and I'm not sure we've got it [consent] right yet actually. (Governance Representative 7, Male)

Many interviewees felt strongly that patients should be provided with verbal (ie, discussions during consultation) and written information (ie, via a patient information leaflet (PIL)) when being offered innovative surgery. In line with this, most governance representatives reported that innovative procedures could only be introduced if an appropriate PIL had been developed (although there was variation as to whether this was reviewed by the committee).

> We will insist that there is a patient guide…an information sheet for the patient…It [PIL] will talk about the relative inexperience of the team, saying that this is new to us, but that we're doing it under conditions where we've got an expert coming to help us, we've been to appropriate training, it's been approved by the organisation as something that we think will be safe, but that it is a new way of doing things. (Governance Representative 24, Male, Trust 9)

However, other trusts had no mandatory requirements for written information or did not review written documentation as part of the application to introduce a new procedure:

Obviously in situations in which the technique might be new then that's (PIL) not something I've specifically addressed but probably should do. (Governance Representative 48, Male, Trust 24)

I definitely think they should know it's a new procedure. The way we probably tell (patients) is only in the consent process verbally, I don't recall seeing that in any of our written leaflets. (Governance Representative 47, Female, Trust 18)

## The challenges of discussing innovation

As previously discussed, surgeons felt a need to deliver additional information during consultations for innovative surgery. However, data suggest that some surgeons may find this challenging:

it's just trying to get across not only this fact that this is what it is, but the fact that it's newer than something else which as a standard we use. That's quite difficult trying to get that across, and some patients will understand better than others. (Surgeon 42, Neurosurgery, Male, Trust 13)

At a practical level, surgeons reported finding it difficult to establish what information is essential and how much detail is enough. These issues may be further compounded when faced with time constraints, which limit surgeons' ability to have long and/or multiple discussions with patients:

We don't have much time …I'm finding that I'm over talking far too much information, but I need to make them aware that they're essentially guinea pigs. (Surgeon 23, Colorectal, Female, Trust 2)

Delivering information to a variety of patients who may differ in their ability to understand complex—and potentially very technical—information was a further challenge. To help overcome this, some surgeons felt it was important to simplify technical information and, when possible, provide alternative mediums of information to aid decision-making:

It's [surgical procedure] such a complex thing to be explaining to people so you have to try and bring it down to simplistic terms but at the same time give it in a way that actually they can process the salient points of the information.' (Surgeon 46, Trauma and Orthopaedic, Male, Trust 22)

I think one of the most important things about when you're making this change is that you give people different forms of access, and of course now it's easy. There are YouTube videos of stuff. (Surgeon 43, Orthopaedic, Male, Trust 2)

Surgeons acknowledged that innovative surgery was associated with uncertainties, including a lack of knowledge surrounding risks, benefits and long-term outcomes. Some felt that discussing uncertainties may conflict with the professional norms within surgery:

This is what surgeons find hard. You have to say, "I don't know," and we hate that. But actually, we should say it more often. We should say it more often. (Surgeon 4, General and Upper Gastrointestinal, Male, Trust 12)

As a result, when faced with uncertainty surrounding an innovative surgery, surgeons sometimes appeared to disclose their personal opinions during patient consultations:

They were told that it was a new technique, that we weren't sure what the long-term outcomes were going to be but, we thought that it was better than traditional surgery. (Surgeon 16, Urology, Male, Trust 2)

On the other hand, some expressed a desire to remain impartial and stressed the importance of providing balanced information. Several interviewees described using strategies to help ensure they remained as objective as possible during consultations. Examples discussed included adhering to scripted information and only discussing innovative surgery if a patient specifically requested an alternative to conventional approaches:

You can word a new procedure, this is amazing, fantastic, amazing, it's phenomenal and there's no risks to it, and all the patients will say sure, yeah sign me up. Whereas, this is what we're trying to look for, this is the benefits, these are the potential risks, so on and so forth. Then it becomes much more objective. (Surgeon 25, Thoracic, Male, Trust 14)

Of note however, the language used to describe conventional surgical approaches during interviews highlights the potential for surgeons—although unintentionally—to influence treatment choice. For instance, terms such as 'old fashioned' and 'traditional' were used by some when describing how they approached patient information provision:

I'll tell them, "We're going to try and do it like this. Is that okay? Do you have any objection? The alternative is we're going to do it the old-fashioned way, which is make a big cut in your- on your stomach wall." (Surgeon 4, General and Upper Gastrointestinal Surgeon, Male, Trust 12)

## Managing patient expectations

Surgeons felt that patients often had preferences for new treatments and expected these to deliver better outcomes. For instance those with experience of introducing robotically assisted surgery commonly reported that patients conveyed a strong preference for this approach. Consistent with this, one interviewee felt that patient demand for robotically assisted surgery was a driving factor for hospitals to invest in robotics.

Patients took to it [robotically assisted surgery] very well. I thought they might be a bit nervous about it, you know the robot taking over or going crazy…patients like the idea of technology. They equate the robot to being an advanced, superior treatment. (Surgeon 31, Colorectal, Male, Trust 8)

Patients wanted to have robotic surgery, and if you didn't have a robot, the patients would just go to a neighbouring hospital that did. So, it's really patient driven, so if you wanted to be a centre you have to have a robot. (Surgeon 16, Urology, Male, Trust 2)

In general, interviewees felt patients held a misconception that 'new' surgical procedures and technologies are better than those normally provided. Some surgeons reported that patients had actively sought them out in order to receive innovative surgery:

They tend to have sought you out, because they want something different than the standard. So they're kind of are willing to accept uncertainty and risk because they've already sought out that uncertainty and risk, if you know what I mean. (Surgeon 9, Upper Gastrointestinal, Male, Trust 3)

Several surgeons were concerned that patients did not understand that innovative procedures may carry uncertainties in terms of risk and benefit. Managing patients' preferences, particularly if they align with the personal preferences of the surgeon, appeared challenging for some:

When does a patient turn around and say, "Well I'll have the old one?" And the answer is never, and that's difficult, because they are immediately signing themselves up for an experiment. And how you describe that to patients is really difficult, because you've got all of your intrinsic biases yourself. Okay, go on then. (Surgeon 43, Orthopaedic, Male, Trust 2)

I don't think I've had anyone say, "No, thank you." I think most of them have gone, "I don't mind being a guinea pig." They always use that phrase (laughter). Makes me feel a little bit uncomfortable. (Surgeon 36, Breast, Female, Trust 4)

## DISCUSSION

This study provides novel insights into surgeons' and governance representatives' views on information provision in the context of innovative surgical procedures and represents a significant contribution to the limited empirical literature. Results highlight inconsistencies in how NHS Trusts govern how patients should be informed about innovative surgical procedures. Furthermore, interviews with surgeons suggest a lack of consistency in amount and quality of the provision of verbal and written information prior to delivery of innovative surgical treatment. Data indicate a heavy reliance on surgeons to decide what and how information is communicated to patients within some trusts. This conflicts with the recommendations published by the RCSEng, which suggest oversight committees should discuss information provision and consent with surgeons prior to approvals being granted.[8] Collectively, our results stress the need for better oversight and regulation of information provision in the context of innovative surgery. Without improved standards of information provision, patients' informed consent is potentially compromised. There is, however, particular complexity in understanding what innovation is, and when it might occur, that makes establishing these standards difficult.

Consistent with previous research, our findings suggest that surgeons may struggle to clearly define surgical innovation,[13] and this may impact on their approach to patient information provision and consent. It can represent a continuum from standard practice through to true innovation. Rogers et al[13] demonstrated that surgeons held no uniform view of surgical innovation, and showed the lack of agreement on the distinction between innovation and research. Our study highlights the implications of ambiguity—surgeons may use their own judgement as to what is innovative and therefore not discuss novel components of an operation with patients. In addition, procedures that were deemed innovative were sometimes introduced as part of routine clinical practice. This meant that information provided to patients was at the discretion of the surgeon, rather than an ethics review board and no peer review of the process. Birchley et al have suggested that instead of focusing on whether something is 'innovative' or not, the potential risks and the ethical appropriateness of modifying surgical practice should be considered to ensure the safe translation of surgical innovation into clinical practice.[29] Work is ongoing to better conceptualise what innovation is and determine when it is happening[49]—although in the meantime we rely on surgeons' views on what is innovative and how that influences their practice of information provision.

Prior research has shown patients do not always feel fully informed about the different treatment options available to them.[50 51] It is likely that the added complexity associated with innovative treatments (eg, unknown risks and benefits, lack of long-term data, limited surgeon experience of this technique) makes discussing treatments options particularly challenging for surgeons. This has an impact on the process of information provision within the context of gaining informed consent to undergo the procedure. As many innovations are undertaken within the context of clinical practice and not research, then the information provision is at the discretion of the surgeon. Study participants recognised the importance of communicating the risks and benefits, treatment alternatives and novel status of a procedure to patients. However, it was less common for interviewees to identify prior training, experience with undertaking the procedure and the evidence base supporting the innovation as important components of consent in information provision. These elements of information provision are recognised as key to gain informed consent.[2 8] The variation observed in this study suggests there is a lack of clarity in guidance for surgeons on what should be disclosed during the content of informed consent discussions in the context of innovative surgery, meaning that some patients are likely to be better informed than others when making treatment decisions. This is a concern given that information provision within the context of informed consent has been linked to patient satisfaction and rates of litigation following surgery.[52–54]

Aligning with previous research, our data suggest surgeons may not disclose their personal experience of performing

a procedure to patients and this is important to patients.[12] In this study few surgeons identified, or reported discussing, surgical experience and/or training as a component of information provision. While we cannot conclude from the current data, the disclosure of this information could lead to conversations regarding the surgical learning curve. Such discussions could help emphasise to patients that a procedure is novel and there are potential uncertainties in terms of risks and benefits. Training programmes have been developed to enhance informed consent in RCTs,[55 56] although none have focused on innovative surgical procedures. Our study suggests that surgeons may require training to raise awareness of the challenges of discussing novel procedures and provide support to overcome these.

To the best of our knowledge, this study is the first to explore healthcare professionals' views of what information should be shared with patients during informed consent consultations in the context of innovative surgery. An important strength of the study is the sample of both surgeons and governance representatives, recruited from various surgical specialties, trust types and geographical locations. This allowed us to explore information provision and consent from different perspectives, and provided important insight into the role trust governance plays in patient information provision and consent. Nonetheless, there are potential limitations to the study that should be acknowledged. We cannot conclude exactly what information is communicated to patients during consultations from interview data alone. Future research should capture what actually happens in these interactions by audio recording clinician–patient consultations. Such methods have been used successfully in RCTs to improve the quality of informed consent and recruitment rates.[56–58] Furthermore, studies exploring patients' views, particularly those who are offered or have received innovative surgery, are critical if we are to understand the type of information patients draw on when making decisions about their care. This work is currently being conducted by our research group as part of a larger research study exploring innovation in surgery.[49] It is hoped that this research will aid the development of 'core information sets' (minimum sets of information that should be discussed with patients), which will ultimately support surgeons to discuss innovation and help catalyse the discussion of issues that are important to individual patients.[59]

In conclusion, the study suggests that information provision for innovative surgery presents a challenge for surgeons. Our findings also highlight variation in how NHS Trusts govern information provision and consent practice for innovative surgery, and further inconsistencies between the views of surgeons and governance representatives. When innovations are introduced under the auspices of research, there are governance structures (eg, research ethics committees) in place to help ensure patient information is appropriate. However, innovation and modifications to surgery often take place without formal research approval, which raises the question of how well informed patients are prior to surgery. While the recent RCSEng guidelines for introducing new techniques and technologies represent an important step

towards improving informed consent, our study suggests a need to develop support and training programmes to ensure that surgeons consistently deliver appropriate and comprehensive information to patients in the context of innovative surgery.

**Author affiliations**
[1]National Institute for Health Research Bristol Biomedical Research Centre Surgical Innovation Theme, University of Bristol, Bristol, UK
[2]Bristol Centre for Surgical Research, Bristol Medical School: Population Health Sciences, University of Bristol, Bristol, UK
[3]Division of Surgery, University Hospitals Bristol NHS Foundation Trust, Bristol, UK
[4]Division of Surgery, University Hospitals Bristol and Weston NHS Foundation Trust, Bristol, UK
[5]GI Surgery, North Bristol NHS Trust, Bristol, UK
[6]Vascular Services, North Bristol NHS Trust, Bristol, UK

**Contributors** JMB conceptualised the idea to study informed consent and information provision in the context of surgical innovation. DE developed the idea for the current study with support from JMB, JZ, SP, SC, NSB, BGM, AGKM, JM and KA. NSB, JM, AGKM and BGM identified potential participants. JZ and DE conducted the interviews and analysed the data, with regular discussion of data interpretation with SP and JMB. The NIHR Bristol Biomedical Research Centre Surgical Innovation theme is directed by JMB (theme lead) and RH (deputy theme lead). The manuscript was written by JZ, and DE critically reviewed the manuscript and responded to reviewers' comments. All authors commented on the article.

**Funding** This study was funded by the NIHR Biomedical Research Centre at University Hospitals Bristol and Weston NHS Foundation Trust and the University of Bristol, with support from the MRC ConDuCT-II (Collaboration and innovation for Difficult and Complex randomised controlled Trials In Invasive procedures) Hub for Trials Methodology Research (MR/K025643/1). BM is an NIHR clinical lecturer, AM is funded by a Clinician Scientist Fellowship (NIHR-CS-2017-17-010) from the National Institute for Health Research, and NB is funded by an MRC Clinician Scientist Award.

**Disclaimer** The views expressed are those of the author(s) and not necessarily those of the NIHR or the MRC.

**Competing interests** None declared.

**Patient consent for publication** Not required.

**Provenance and peer review** Not commissioned; externally peer reviewed.

**Data availability statement** No data are available. All data relevant to the study are included in the article or uploaded as supplemental information.

**ORCID iDs**
Jesmond Zahra http://orcid.org/0000-0002-7947-2216
Natalie S Blencowe http://orcid.org/0000-0002-6111-2175
Kerry Avery http://orcid.org/0000-0001-5477-2418
Angus G K McNair http://orcid.org/0000-0002-2601-9258

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
