## [Reviewer comments · BMJ Open]

ARTICLE DETAILS

TITLE (PROVISIONAL)	Discussing surgical innovation with patients: A qualitative study of surgeons' and governance representatives' views
AUTHORS	Zahra, Jesmond; Paramasivan, Sangeetha; Blencowe, Natalie; Cousins, Sian; Avery, Kerry; Mathews, Johnny; Main, Barry; McNair, Angus; Hinchliffe, Robert; Blazeby, Jane; Elliott, Daisy

VERSION 1 – REVIEW

REVIEWER	Margaret M Byrne Moffitt Cancer Center Tampa FL USA
REVIEW RETURNED	11-Dec-2019

GENERAL COMMENTS	Review of: bmjopen-2019-035251 Title: Informed consent in the context of surgical innovation: A qualitative study of stakeholder views, Journal: BMJ Open Reviewer: MM Byrne Comments for authors: This manuscript reports results of a qualitative study which collected information from both physicians and government representative on the topic of innovative surgical procedures. Although this topic is an interesting and potentially important topic, and the investigators /manuscript have several laudable facets (noted below), there are also a number of concerns which substantially reduce my enthusiasm for the manuscript as written. These concerns are described first below, and are somewhat ordered by importance. Laudatory elements of the manuscript follow. Concerns 1. "Innovative surgery" compared to "standard surgery" Perhaps this is a philosophical or semantic quibble, but I don't think so. I was unhappy with the comparison throughout the manuscript of "innovative" versus "standard" surgery and so how the informed consent process for innovative surgery must be quite different from "standard" surgery. For example, much is made of the fact that there are uncertainties in risks and benefits in innovative surgery – and thus the need for good communication between physicians and patients for good informed consent.
---

BUT, this is exactly the same in the situation of getting informed consent for standard surgery! We do not know with 100% certainty what are the risks and benefits, and outcomes/successes, of ANY surgery. There might be smaller confidence intervals around risks, benefits, and outcomes of standard surgery, but uncertainty and the need to educate patient about it, is still present.

Here is an excerpt from the narrative, which is an example of the type of narrative/comparison that I object to:
"There was a consensus both across and within participant groups that if a procedure is deemed to be innovative, patients should be provided with additional information extending beyond that given during routine surgical consultations. The potential risks and benefits, alternative treatment options, and the novel status of the procedure were universally regarded as essential information..."
Does this statement mean that the participants – or investigators – believed that individuals getting a "routine" surgical procedure do NOT have to be told about risks and benefits or alternative treatments? I really hope not. But it is instances like this that reduce my enthusiasm for this presentation of this study.

To this reviewer, the current manuscript sets up innovative surgery as something that is qualitatively different from standard surgery. However, this seems to me to be an inappropriate way to think about it. And there is justification for this view even in the narrative itself. For example, one of the included quotes discusses that it is really a continuum.

Thus, to me the counter positioning of standard vs innovative is a mistake and distraction. Instead, I would see all surgeries (and any type of medical intervention actually), as on a continuum of "familiarity" where at one end there are interventions such as cataract surgery (thousands done every year, and quite precise information on the likelihood of success or different side effects). And at the other end are things like the Concorde Lift Expandable Interbody Device.

Therefore, the idea of a continuum rather than "Comparison of different types of things" might make more sense.

2. "Stakeholders"

The results presented here include only information from physicians and "government representative". Because patients or potential patients are not included in this study (or at least the data presented here), there is a gaping hole in the information on the topic of discussions of innovative surgical procedures. Without the patient view, it is much harder to find or describe how this research will contribute to the overall issue of informing patients about innovative surgical procedures.

Therefore, if the overall study did also collection information from patients – who in my mind are the most important "stakeholders" in this situation – I would strongly encourage the investigators to include those data in this manuscript.

3. Title & Relationship to Informed Consent:

	Because the title uses the phrase "Informed consent", I was expecting that the results would in fact say something about informed consent. But instead, although the narrative lightly links the information received from the "stakeholders" to the informed consent process, this is really NOT a study of informed consent as it is so narrowly focused and doesn't yield any specific insights that can help informed consent. 4. Conclusions in the abstract As written, the "Conclusion" section of the abstract seems to indicate that there were NO useful findings from the research described here. I would suggest making sure that there are clear statements in the conclusions about what was done and why it is important. 5. Recruitment and participants In the methods section on Sampling and recruitment, clarification is needed with respect to the number of potential participants who were invited to take part (n=83, according to the text), vs. the number who were interviewed (n=42). Of the 41 who were invited but did not participate, how many declined and why? How many were "dropped" by the study because saturation had been reached? Laurels 1. Methods: Data analysis The process – and description of the process – for analyzing the qualitative data was well described and complete. Too many qualitative studies have very weak descriptions of their analyses, or claim that they are using a certain analysis procedure ("grounded analysis" is a huge one), when in fact they are not. 2. Adaptation of the interview guides/schedules I applaud the methodology of allowing the interview schedule to develop and be modified based on the information collected (and overall tenor) of the initial interviews.
--	---

REVIEWER	Cynthia Kendell Nova Scotia Health Authority, Canada
REVIEW RETURNED	20-Dec-2019

GENERAL COMMENTS	Thank you for the opportunity to review this manuscript. This paper addresses a topic area that has important clinical, legal, and ethical implications—that of informed consent. The submitted manuscript is promising, but I believe it would be strengthened by a number of revisions. I have provided my feedback below and expect that the majority of recommendations will be relatively easy to incorporate, although some will require substantial revisions. Overall, the main things that I feel need to be addressed are as follows:
--

- 1) Since this paper is about informed consent, the authors should state the current legal and ethical requirements for informed consent in the UK at the outset of the paper.
- 2) There is no clearly stated question or objective to anchor the paper.
- 3) Methodology has not been addressed and methods require support from the literature.
- 4) It is not clear how innovations that are introduced via research channels (e.g. clinical trials) fit into this study. Once the authors have had an opportunity to address this feedback, I would gladly review the article again.

Abstract

- The authors state that 26 clinicians participated in this study. If all 26 clinicians were surgeons, please replace "clinician" with "surgeon".

Summary of Strengths and Limitations

- Bullet #2- Replace "patient" with "patients".

Introduction

- Para 1, Sentence 2- Provide a citation for the Montgomery ruling.
- Para 1, General Comment- Since the legal standards for informed consent are of great importance in the context of this manuscript, I recommend that the authors use the first paragraph to clearly set out the current legal requirements for informed consent. Specifically, I recommend clearly explaining that the previous standard (the Bolam test) focused on whether the physician had acted in line with responsible body of opinion, whereas the Montgomery ruling set a new standard whereby physicians must ensure that patients are made aware of any material risks (i.e., risks that a reasonable person in the patient's position would find significant). This gives the reader a much-needed point of reference in terms of what physicians should be doing.
- Para 2, Sentence 6- Please revise this sentence for improved accuracy. Suggested wording: "Recognizing these issues, the RESEng has recently published guidelines for the development, implementation and dissemination of surgical innovation, which contains guidance on informed consent".
- Para 2, Sentence 7- The listed items do appear in the guidance document that is cited, however, the original citation for these items is a review by Broekman et al. Suggesting wording: "This document suggests, based on findings of a review by Broekman et al citation, that patients should be informed of.....".
- Para 3, Sentence 1- The authors state that "...consent in early phase surgical studies or when new or modified surgical treatments are introduced as part of routine clinical practice (e.g., via trust governance processes) has not previously been explored." However, there seems to be a rather substantial body of literature that exploring consent relevant to surgical innovations, as evidenced by the content analysis (Bracken-Roche et al) and other articles (Reitsma and Moreno; Char et al) that are cited by the authors. Please clarify.
- Para 3, Sentence 4- The authors refer to a "scarcity of research in this area". What area exactly? There is literature on informed consent in the context of surgical innovation. Is it that the perspectives of stakeholders on how to approach informed consent has not been studied? Please be more explicit with regard to the knowledge gap that is being addressed.

	 • Para 4, Sentence 1- Similar to the previous comment, the authors write “To address this evidence gap...”. Which gap? Please revise paragraph 3 to improve clarity regarding the evidence gap that is being addressed. • Para 4-The specific research objective(s) and/or research question(s) being addressed by this study is unclear. Is the objective to explore how surgical innovations are currently being introduced and discussed with patients in the UK? Please explicitly state. Methods  • General Comment- What is the methodology that is being used? The type of qualitative approach, relevant theory/frameworks, and research paradigm have not been addressed. • General Comment- The methods described are indeed commonly used in qualitative research, however, the methods section is lacking appropriate citations to demonstrate that the approach is based on established practices. The only citation in the entire methods section is for Braun and Clarke, which is a beginner’s guide that is geared toward students. This section would be strengthened by the addition of appropriate references (for sampling strategies, use of semi-structured interviews, interview guide design, analytic approach etc.). • Design, Para 1, Sentence 3- Revise to state, “The study is reported in line with the Standards for Reporting Qualitative Research (SRQR) citation.” • Sampling and Recruitment, Para 1, Sentence 5- The authors mention that individuals with experience in “first in human procedures” were included in the study. This indicates that the interviews with participants also included conversations about innovations introduced as part of clinical trials. This may have influenced study findings substantially. For example, interventions introduced as part of a clinical trials may be more readily identified by surgeons as “innovative”. Moreover, given that consent to participate in research is typically subject to unique regulatory requirements and processes (e.g., REB approval, use specific forms, etc.), it is likely that informed consent requirements in a research-context are more clearly set out compared to a non-research context. Please provide information about whether this study encompassed responses pertaining to surgical innovations introduced as research, versus those being introduced solely as clinical care (i.e., no research component), and the implications for study findings. • Data Collection- The authors have identified two distinct stakeholder groups with whom interviews were conducted: (1) surgeons, and (2) governance representatives. These two groups of individuals would have very different experiences with and perspectives on informed consent in the context of surgical innovation. Please explain why each of these two stakeholder groups were selected and what information you hoped to gain from each groups (e.g., “Governance representatives were included in this study because...”). • Data Collection- The authors explain that the interview guide was developed iteratively. How did the initial interview guide change over the course of the study? • Data Collection- Related to the previous comment, the interview guide that was provided as Appendix 1 seems much more geared toward surgeons. Please provide details about how the questions/focus of the interviews differed between the two groups. How was it adapted for governance representatives?
--	--

	 • Data Analysis- Were transcripts anonymized, or de-identified? If anonymized, please explain process of anonymization. Results  • General Comment-Consider reframing the first two themes so that they are not worded as questions. • General comment –The very first paragraph provides details on study participants, as well as information on data collection. Rather than being placed in the results section, I suggest that this information be placed in the methods section under the appropriate sub-heading (i.e., sampling and recruitment, or data collection data collection). • “What is Innovation?”, fifth quote- The fifth quote under this theme seems to bring up research again (“ ...if you’re not trialling the instrument to see if it’s safe...”). The quote speaks to recognition that the standards for informed consent differ when the innovation is being introduced as part of clinical trial. The difference in context (research vs. non-research) needs to be addressed by the authors in the manuscript. How do innovations introduced via research channels fit in to this study. • “What should patients be told?”, paragraph 2- The authors identify several types of information (evidence, experience, training, safety precautions, national guidance) as not frequently discussed by physicians, but use quotes that state that experience and evidence are in fact discussed by physicians. I would suggest selecting different quotes to better illustrate the point being made (i.e., quotes illustrating that these things are not frequently discussed). Discussion  • Para 2, Sentence 3- Change to “innovative treatments” • Para 5- This is the first time that the authors explicitly address research. The difference in how research is regulated compared to practice changes occurring outside of the scope of research are very different. This should be addressed at the outset of the paper, and warrants substantial attention in the discussion. Re: Standards for reporting qualitative research  • Purpose or research question-Not explicitly stated. • Qualitative approach and research paradigm-Not addressed. • Researcher characteristics and reflexivity-Not addressed.
--	--

VERSION 1 – AUTHOR RESPONSE

Reviewer 1

Review of: bmjopen-2019-035251

Title: Informed consent in the context of surgical innovation: A qualitative study of stakeholder views,

Journal: BMJ Open

Reviewer: MM Byrne

Comments for authors: This manuscript reports results of a qualitative study which collected information from both physicians and government representative on the topic of innovative surgical procedures. Although this topic is an interesting and potentially important topic, and the investigators

/manuscript have several laudable facets (noted below), there are also a number of concerns which substantially reduce my enthusiasm for the manuscript as written.

These concerns are described first below, and are somewhat ordered by importance. Laudatory elements of the manuscript follow.

Concerns

1. "Innovative surgery" compared to "standard surgery"

Perhaps this is a philosophical or semantic quibble, but I don't think so. I was unhappy with the comparison throughout the manuscript of "innovative" versus "standard" surgery and so how the informed consent process for innovative surgery must be quite different from "standard" surgery. For example, much is made of the fact that there are uncertainties in risks and benefits in innovative surgery – and thus the need for good communication between physicians and patients for good informed consent. BUT, this is exactly the same in the situation of getting informed consent for standard surgery! We do not know with 100% certainty what are the risks and benefits, and outcomes/successes, of ANY surgery. There might be smaller confidence intervals around risks, benefits, and outcomes of standard surgery, but uncertainty and the need to educate patient about it, is still present. Here is an excerpt from the narrative, which is an example of the type of narrative/ comparison that I object to: "There was a consensus both across and within participant groups that if a procedure is deemed to be innovative, patients should be provided with additional information extending beyond that given during routine surgical consultations. The potential risks and benefits, alternative treatment options, and the novel status of the procedure were universally regarded as essential information..." Does this statement mean that the participants – or investigators – believed that individuals getting a "routine" surgical procedure do NOT have to be told about risks and benefits or alternative treatments? I really hope not. But it is instances like this that reduce my enthusiasm for this presentation of this study.

To this reviewer, the current manuscript sets up innovative surgery as something that is qualitatively different from standard surgery. However, this seems to me to be an inappropriate way to think about it. And there is justification for this view even in the narrative itself. For example, one of the included quotes discusses that it is really a continuum. Thus, to me the counter positioning of standard vs innovative is a mistake and distraction. Instead, I would see all surgeries (and any type of medical intervention actually), as on a continuum of "familiarity" where at one end there are interventions such as cataract surgery (thousands done every year, and quite precise information on the likelihood of success or different side effects). And at the other end are things like the Concorde Lift Expandable Interbody Device. Therefore, the idea of a continuum rather than "Comparison of different types of things" might make more sense.

Response: We thank the reviewer for highlighting these critical points. It is absolutely correct that for standard surgery it is the responsibility of the surgeon to discuss with patients the risks and benefits and the treatment alternatives we were not proposing otherwise. This would be in keeping professional guidance for information provision for informed consent. We also agree with the point made that there is a spectrum and continuum of standard practice through to true innovation. We do think, however, that at the 'innovation end' of this spectrum the process of informed consent and information provision differs to that required for more routine procedures because of the additional 'unknown unknowns'. We have amended the methods section and the discussion to clarify these points and to include the helpful comments from the reviewer.

Changes to the methods section (page 6):

As there is no standardised definition for innovation (14, 26, 27), and because surgical innovation is unlikely to involve a single discrete development and it is often a continuum from standard practice through to true innovation (28, 29), participants were asked to reflect what the term meant to them and describe procedures that they deemed to be innovative.

Changes to the discussion section (page 16):

Consistent with previous research, our findings suggest that surgeons may struggle to clearly define surgical innovation(50), and this may impact upon their approach to patient information provision and consent. It can represent a continuum from standard practice through to true innovation. Rogers et al (14) demonstrated that surgeons held no uniform view of surgical innovation and showed the lack of agreement on the distinction between innovation and research. Our study highlights the implications of ambiguity – surgeons may use their own judgement as to what is innovative and therefore not discuss novel components of an operation with patients. In addition, procedures that were deemed innovative were sometimes introduced as part of routine clinical practice. This meant that patient information provided was at the discretion of the surgeon, rather than an ethics review board and no peer review of the process. Birchley et al have suggested that instead of focussing on whether something is 'innovative' or not, the potential risks and the ethical appropriateness of modifying surgical practice should be considered to ensure the safe translation of surgical innovation into clinical practice(29). Work is ongoing to better conceptualise what innovation is and determine when it is happening (51) - although in the meantime we rely on surgeons' views on what is innovative and how that influences their practice of information provision.

2. "Stakeholders"

The results presented here include only information from physicians and "government representative". Because patients or potential patients are not included in this study (or at least the data presented here), there is a gaping hole in the information on the topic of discussions of innovative surgical procedures. Without the patient view, it is much harder to find or describe how this research will contribute to the overall issue of informing patients about innovative surgical procedures. Therefore, if the overall study did also collection information from patients – who in my mind are the most important "stakeholders" in this situation – I would strongly encourage the investigators to include those data in this manuscript.

Response: We completely agree that exploring patient views are very important. This is part of the next phase of the study and it will inform a separate report.

We have now acknowledged this as a key limitation in the strengths and imitations summary section, as well as the abstract conclusion (pages 2/3/4):

'Further work should capture patients' views on information provision in this context, so that recommendations can be made to improve transparency and communication.'

'Further work should capture consultations where new procedures are discussed with patients and patients' views of these information exchanges.'

3. Title & Relationship to Informed Consent:

Because the title uses the phrase "Informed consent", I was expecting that the results would in fact say something about informed consent. But instead, although the narrative lightly links the information received from the "stakeholders" to the informed consent process, this is really NOT a study of informed consent as it is so narrowly focused and doesn't yield any specific insights that can help informed consent.

Response: We agree this should have been clearer and the focus if more on information provision than informed consent (although they are linked). We have now changed the title of the manuscript and updating the discussion to include detail on the relationship between information provision and informed consent:

New title of manuscript:

"Discussing surgical innovation with patients: A qualitative study of surgeons' and governance representatives' views"

Changes to the discussion section (pages 16/17). We also updated the introduction on page 5 – see response to the second reviewer below (page 6 of this document):

Prior research has shown patients do not always feel fully informed about the different treatment options available to them(52, 53). It is likely that the added complexity associated with innovative treatments (e.g. unknown unknown risks and benefits, lack of long-term data, limited surgeon experience of the technique) makes discussing treatments options particularly challenging for surgeons. This has an impact on the process of information provision within the context of gaining informed consent to undergo the procedure. As many innovations are undertaken within the context of clinical practice and not research, then the information provision is at the discretion of the surgeon. Study participants recognised the importance of communicating the risks and benefits, treatment alternatives, and novel status of a procedure to patients. However, it was less common for interviewees to identify prior training, experience with undertaking the procedure, and the evidence base supporting the innovation as important components of consent in information provision. These elements of information provision are recognised as key to gain informed consent (2,9). The variation observed in this study suggests there is a lack of clarity in guidance for surgeons on what should be disclosed during the content of informed consent discussions in the context of innovative surgery, meaning that some patients are likely to be better informed than others when making treatment decisions. This is a concern given that information provision within the context of

informed consent has been linked to patient satisfaction and rates of litigation following surgery(54-56).

Aligning with previous research, our data suggests surgeons may not disclose their personal experience of performing a procedure to patients and this is important to patients (57). In this study few surgeons identified, or reported discussing, surgical experience and/or training as a component of information provision. While we cannot conclude from the current data, the disclosure of this information could lead to conversations regarding the surgical learning curve such discussions could help emphasise to patients that a procedure is novel and there are potential uncertainties in terms of risks and benefits.

4. Conclusions in the abstract

As written, the "Conclusion" section of the abstract seems to indicate that there were NO useful findings from the research described here. I would suggest making sure that there are clear statements in the conclusions about what was done and why it is important.

Response: Thank you for pointing this out. We have rewritten the conclusion in the abstract to specifically address the implications on informed consent and information provision (pages 2/3):

This study highlights the challenges of information provision and obtaining informed consent in the context of innovative surgery, including establishing if and how a procedure is truly innovative, determining the key information to discuss with patients, ensuring information provision is objective and balanced, and managing patient expectations and preferences. This suggests that surgeons may require support and training to discussing novel procedures with patients. Further work should capture consultations where new procedures are discussed with patients and patients' views of these information exchanges.

5. Recruitment and participants

In the methods section on Sampling and recruitment, clarification is needed with respect to the number of potential participants who were invited to take part (n=83, according to the text), vs. the number who were interviewed (n=42). Of the 41 who were invited but did not participate, how many declined and why? How many were "dropped" by the study because saturation had been reached?

Response: We apologise that this was unclear. We have now added this information on Page 7 of the methods (page 9):

Of the 83 participants that were approached, 42 participants were recruited (those who did not participate were unable to find time for an interview or did not respond to the researcher).

1. Methods: Data analysis

The process – and description of the process – for analyzing the qualitative data was well described and complete. Too many qualitative studies have very weak descriptions of their analyses, or claim that they are using a certain analysis procedure ("grounded analysis" is a huge one), when in fact they are not.

Response: Thank you for this positive feedback.

2. Adaptation of the interview guides/schedules

I applaud the methodology of allowing the interview schedule to develop and be modified based on the information collected (and overall tenor) of the initial interviews.

Response: Thank you for your encouraging comment.

Reviewer 2: Comments

Thank you for the opportunity to review this manuscript. This paper addresses a topic area that has important clinical, legal, and ethical implications—that of informed consent. The submitted manuscript is promising, but I believe it would be strengthened by a number of revisions. I have provided my feedback below and expect that the majority of recommendations will be relatively easy to incorporate, although some will require substantial revisions. Overall, the main things that I feel need to be

addressed are as follows:

1) Since this paper is about informed consent, the authors should state the current legal and ethical requirements for informed consent in the UK at the outset of the paper.

Response: Thank you for making this important point. We have now added an overview of the legal requirements (page 5):

Until recently, the Bolam ruling meant that a doctor's duty to inform patients prior to surgery was used to judge on whether they had acted in line with the view of a responsible body of medical opinion (1, 2). The 2015 Montgomery ruling(3) redefined the standard of informed consent in the United Kingdom (UK) and represented a shift from a paternalistic model of consent to a more patient-centred approach(4, 5) – meaning that a patient should be told whatever they want to know, not what the doctor thinks they should be told(6).

2) There is no clearly stated question or objective to anchor the paper.

Response: Thank you for pointing this out. We have now added the following to the introduction (page 6):

The current study sought to explore surgeons' and governance representatives' views of information provision and informed consent for the introduction of new invasive procedures in the United Kingdom (UK).

3) Methodology has not been addressed and methods require support from the literature.

Response: We apologise about not including more detail and feel this is an important point. We have now added the following paragraph to the analysis section on page 8:

Interview recordings were transcribed and de-identified so that names and identifiable information were removed. Thematic analysis was undertaken using the constant comparison technique of grounded theory (41, 42), which involved using the inductive identification of codes from the data to generate new hypotheses about phenomena that are derived or grounded in the data (43). Its central principle is of constant comparison, where new findings are systematically compared with existing data so that similarities and differences can be identified and emerging theories refined through the ongoing assimilation of data (41, 44).

We have also added the references throughout the methods section so that the following references are included:

O'Brien, B.C., et al., Standards for reporting qualitative research: a synthesis of recommendations. Acad Med, 2014. 89(9): p. 1245-51.

Willig, C., Applied discourse analysis: Social and psychological interventions. 1999: Open University Press.

Burman, E., Minding the Gap: Positivism, Psychology, and the Politics of Qualitative Methods. 1997. 53(4): p. 785-801.

Dempster, M., A Research Guide for Health and Clinical Psychology. 2011: Palgrave Macmillan.

Patton, M.Q.J.Q.s.w., Two decades of developments in qualitative inquiry: A personal, experiential perspective. 2002. 1(3): p. 261-283.

Fassinger, R.E.J.J.o.c.p., Paradigms, praxis, problems, and promise: Grounded theory in counseling psychology research. 2005. 52(2): p. 156.

- Sandelowski, M.J.R.i.n. and health, *Sample size in qualitative research*. 1995. 18(2): p. 179-183.
- Braun, V. and V. Clarke, *Successful qualitative research: A practical guide for beginners*. 2013: sage.
- Bowen, G.A., *Naturalistic inquiry and the saturation concept: a research note*. 2008. 8(1): p. 137-152.
- Rubin, H.J. and I.S. Rubin, *Qualitative interviewing: The art of hearing data*. 2011: sage.
- Hammersley, P. and M. Atkinson, *Ethnography: principles and practice*. 1983, London: Routledge.
- Smith, J.A.J.R.m.i.p., *Semi-structured interviewing and qualitative analysis*. 1995.
- Britten, N.J.B., *Qualitative research: qualitative interviews in medical research*. 1995. 311(6999): p. 251-253.
- Brinkmann, S. and S. Kvale, *Interviews: Learning the craft of qualitative research interviewing*. Vol. 3. 2015: Sage Thousand Oaks, CA.
- Fielding, N. and H. Thomas, *Qualitative interviewing*. 2008.
- Charmaz, K. and L.J.T.S.h.o.i.r.T.c.o.t.c. Belgrave, *Qualitative interviewing and grounded theory analysis*. 2012. 2: p. 347-365.
- Glaser, B.G. and A.L. Strauss, *The Discovery of Grounded Theory*. 1967, Aldine Transaction: USA.
- Seale, C. and D.J.T.E.J.o.P.H. Silverman, *Ensuring rigour in qualitative research*. 1997. 7(4): p. 379-384.
- Krippendorff, K.J.H.c.r., *Reliability in content analysis: Some common misconceptions and recommendations*. 2004. 30(3): p. 411-433.
- Miles, M.B., et al., *Qualitative data analysis: An expanded sourcebook*. 1994: sage.
- Pope, C., S. Ziebland, and N. Mays, *Analysing qualitative data*. *BMJ*, 2000. 320(7227): p. 114-116.
- Braun, V. and V. Clarke, *Using thematic analysis in psychology*. *Qualitative Research in Psychology*, 2006. 3(2): p. 77-101.
- Charmaz, K., *Constructing grounded theory: A practical guide through qualitative analysis*. 2006: sage.
- Barbour, R.S.J.B., *Checklists for improving rigour in qualitative research: a case of the tail wagging the dog?* 2001. 322(7294): p. 1115-1117.
- Dey, I., *Qualitative data analysis: A user friendly guide for social scientists*. 2003: Routledge.

4) It is not clear how innovations that are introduced via research channels (e.g. clinical trials) fit into this study.

Response: Thank you for pointing this out, we agree that this provides important context for the findings and have added the following to the results section (page 10), as well as adding a new table (Table 2) showing participants' experiences of governance for new procedures:

There was variation in how the procedure had been introduced in participants' hospitals. Within this, surgeons' own experiences included requesting approval from their trust's new procedure committee, conducting the procedure in the context of a research study or deeming that approval was not necessary as the procedure was a variation of an existing one (from either judgement of the surgeon or confirmation from a committee). Table 2 highlights that how the challenges of defining innovation, alongside the required technical knowledge of the procedure, could cause uncertainty as to what governance was required:

"So, I'm not convinced that every innovative thing in the trust goes through the committee, because a lot of it depends on the diligence of the people who are introducing new things. So, it's not a bad process but it's just whether it's comprehensive enough, I guess." (Surgeon 1, Male, Neurology, Trust 1)

Table 1: Participants' experiences of governance for new procedures

	Example quote
Procedure considered and approved by trust's new procedure committee	"There's one surgeon in [European country] who described the procedure, and the surgeon in [Asian country] who had done some of these procedures, but not really clear on how they'd done it. I contacted the [European] surgeon. He gave me some pointers, so I discussed it with my colleagues, and of course put it through the governance processes as a novel and a new procedure. And then we did it." (Surgeon 31, Colorectal, Male, Trust 8)
Approval from trust's new procedure committee deemed unnecessary as felt to be a variation of established procedure	"We asked about it to the new procedures committee, but they said well it's basically [variation of established procedure] isn't it. The technicalities of it are lost on non-thoracic surgeons really, which is good." (Surgeon 31, General surgery, Male, Trust 8) "There's a new technique that's probably done by no more than 9 or 10 surgeons in the country. Again, it's a new technique, it's only been reported in the last 3 or 4 years. [...] Because it was under the umbrella of [broader procedure], I didn't have to go through any, so within the hospital there's a board about new techniques, you have to sort of, clinical effectiveness group, you have to have that approved. But, because this was a variation, we didn't need that approval." (Surgeon 22, Male, Orthopedics, Trust 7)

Procedure evaluated in the context of a research study

*“We were looking [procedure], and so that was in the guise of a randomised double-blind trial that we put through Ethics. So that would be innovative. We’ve got an Ethics Committee, and as part of that if you were going to be doing something really brand new that would have to be conducted as part of a trial, so, but when there’s a blur between doing something slightly different then no I don’t think it’s the place for that [...] I can’t think of anything where they’ve been involved in, you know, anything that we’re doing.”
(Surgeon 46, Male, Orthopedics, Trust 22)*

Once the authors have had an opportunity to address this feedback, I would gladly review the article again.

Abstract

- The authors state that 26 clinicians participated in this study. If all 26 clinicians were surgeons, please replace “clinician” with “surgeon”.

Response: We agree this is much clearer. This has now been updated throughout the article.

Summary of Strengths and Limitations

- Bullet #2- Replace “patient” with “patients”.

Response: This section has been updated.

Introduction

- Para 1, Sentence 2-Provide a citation for the Montgomery ruling.

Response: We have now added this citation (in bold, Page 5):

The 2015 Montgomery ruling(3) redefined the standard of informed consent in the United Kingdom (UK).

- Para 1, General Comment-Since the legal standards for informed consent are of great importance in the context of this manuscript, I recommend that the authors use the first paragraph to clearly set out the current legal requirements for informed consent. Specifically, I recommend clearly explaining that the previous standard (the Bolam test) focused on whether the physician had acted in line with responsible body of opinion, whereas the Montgomery ruling set a new standard whereby physicians must ensure that patients are made aware of

any material risks (i.e., risks that a reasonable person in the patient's position would find significant). This gives the reader a much-needed point of reference in terms of what physicians should be doing.

Response: We have now rewritten this section so that it includes an overview of how the legal requirements have changed over time (page 5):

Until recently, the Bolam ruling meant that a doctor's duty to inform patients prior to surgery was used to judge on whether they had acted in line with the view of a responsible body of medical opinion (1, 2). The 2015 Montgomery ruling(3) redefined the standard of informed consent in the United Kingdom (UK) and represented a shift from a paternalistic model of consent to a more patient-centred approach(4, 5) – meaning that a patient should be told whatever they want to know, not what the doctor thinks they should be told(6). (Page 5)

- Para 2, Sentence 6- Please revise this sentence for improved accuracy. Suggested wording: "Recognizing these issues, the RESEng has recently published guidelines for the development, implementation and dissemination of surgical innovation, which contains guidance on informed consent".

Response: Thank you for your suggestion – this has now been updated in the introduction (page 5).

- Para 2, Sentence 7- The listed items do appear in the guidance document that is cited, however, the original citation for these items is a review by Broekman et al. Suggesting wording: "This document suggests, based on findings of a review by Broekman et al citation, that patients should be informed of.....".

Response: We agree this is a much clearer way of phrasing this and have updated the text with your suggestion on page 5:

This suggests, based on findings of a review by Broekman et al (9), that patients should be informed of the innovative nature of the procedure, surgeons' experience with the procedure and the learning curve, the risk and benefits – including unknown or unforeseeable risks or outcomes, the evidence or lack thereof, and alternatives to the innovative procedure.

- Para 3, Sentence 1- The authors state that "...consent in early phase surgical studies or when new or modified surgical treatments are introduced as part of routine clinical practice (e.g., via trust governance processes) has not previously been explored." However, there seems to be a rather substantial body of literature that exploring consent relevant to surgical innovations, as evidenced by the content analysis (Bracken-Roche et al) and other articles (Reitsma and Moreno; Char et al) that are cited by the authors. Please clarify.

Response: We apologise for the confusion. The Bracken-Roche study was a systematic

content analysis of the conceptual literature, which highlighted the ethical issues of obtaining informed consent for innovative procedures. The Reitsma et al and Char et al studies were both quantitative surveys. The Reitsma study asked surgeons about their current practice for discussing new procedures with patients, whereas the Char study asked surgeons and hypothetical patients what information was important to disclose for new procedures. We have now rewritten the introduction so that it elaborates on what these studies did (page 5):

Reviews and commentaries have highlighted potential issues around informed consent for innovative surgical procedures, including patients' and surgeons' beliefs that new treatments constitute improved treatment options or result in better outcomes (10, 11). However, little is known about current practice. Surveys that have been conducted in the United States suggest that information varies considerably in content and quality. Reitsma and Moreno (2002) reported that of twenty-one surgeons, 75% verbally informed patients about the innovative nature of the procedure but only 33% provided written information (12). Char and colleagues (2013), in a scenario based survey carried out with 85 surgeons and 383 patients, found differences between what surgeons thought patients wanted to know about innovative procedures and what patients reported they wanted to know. For instance, compared with surgeons, patients placed more importance on nearly all types of information, particularly volumes and outcomes (13).

- Para 3, Sentence 4- The authors refer to a “scarcity of research in this area”. What area exactly?

Response: We agree this was too vague. We have now added two paragraphs into the introduction section explaining what previous research did and explained how our research builds on this (pages 5 and 6):

Reviews and commentaries have highlighted potential issues around informed consent for innovative surgical procedures, including patients' and surgeons' beliefs that new treatments constitute improved treatment options or result in better outcomes (10, 11). However, little is known about current practice. Surveys that have been conducted in the United States suggest that information varies considerably in content and quality. Reitsma and Moreno (2002) reported that of twenty-one surgeons, 75% verbally informed patients about the innovative nature of the procedure but only 33% provided written information (12). Char and colleagues (2013), in a scenario based survey carried out with 85 surgeons and 383 patients, found differences between what surgeons thought patients wanted to know about innovative procedures and what patients reported they wanted to know. For instance, compared with surgeons, patients placed more importance on nearly all types of information, particularly volumes and outcomes (13).

Whilst a handful of qualitative studies have been conducted exploring how innovative procedures have been introduced (14) (15) (16) (17), none have specifically looked at information provision and informed consent. Moreover, these studies were conducted in Australia and Canada and findings may not be generalisable to other countries and different healthcare systems(18). New invasive procedures may be introduced in the context of formal research studies or via local hospital policies (19). The National Institute for Health and Care

Excellence (NICE) recommends that local NHS organisations have appropriate governance structures in place to review, approve and monitor the introduction of new invasive procedures (20). The current study sought to explore surgeons' and governance representatives' views of information provision and informed consent for the introduction of new invasive procedures in the United Kingdom (UK).

- There is literature on informed consent in the context of surgical innovation. Is it that the perspectives of stakeholders on how to approach informed consent has not been studied? Please be more explicit with regard to the knowledge gap that is being addressed.

We have now also added a sentence in the strength and limitations summary box, which explains how this study is unique (page 4):

This is the first qualitative study to conduct in-depth, semi structured interviews to understand current practice for information provision and informed consent for innovative surgical procedures in the United Kingdom.

- Para 4, Sentence 1- Similar to the previous comment, the authors write "To address this evidence gap...". Which gap? Please revise paragraph 3 to improve clarity regarding the evidence gap that is being addressed.

Response: As previously described, we have rewritten the introduction so that it is clearer how the study builds on previous research (page 5 and 6):

Reviews and commentaries have highlighted potential issues around informed consent for innovative surgical procedures, including patients' and surgeons' beliefs that new treatments constitute improved treatment options or result in better outcomes (10, 11). However, little is known about current practice. Surveys that have been conducted in the United States suggest that information varies considerably in content and quality. Reitsma and Moreno (2002) reported that of twenty-one surgeons, 75% verbally informed patients about the innovative nature of the procedure but only 33% provided written information (12). Char and colleagues (2013), in a scenario based survey carried out with 85 surgeons and 383 patients, found differences between what surgeons thought patients wanted to know about innovative procedures and what patients reported they wanted to know. For instance, compared with surgeons, patients placed more importance on nearly all types of information, particularly volumes and outcomes (13).

Whilst a handful of qualitative studies have been conducted exploring how innovative procedures have been introduced (14) (15) (16) (17), none have specifically looked at information provision and informed consent. Moreover, these studies were conducted in Australia and Canada and findings may not be generalisable to other countries and different healthcare systems(18). New invasive procedures may be introduced in the context of formal research studies or via local hospital policies (19). The National Institute for Health and Care Excellence (NICE) recommends that local NHS organisations have appropriate governance

structures in place to review, approve and monitor the introduction of new invasive procedures (20). The current study sought to explore surgeons' and governance representatives' views of information provision and informed consent for the introduction of new invasive procedures in the United Kingdom (UK).

- Para 4-The specific research objective(s) and/or research question(s) being addressed by this study is unclear. Is the objective to explore how surgical innovations are currently being introduced and discussed with patients in the UK? Please explicitly state.

Response: Thank you for highlighting this, we have now added a specific objective to the introduction section as also explained above (page 6):

The current study sought to explore surgeons' and governance representatives' views of information provision and informed consent for the introduction of new invasive procedures in the United Kingdom (UK).

Methods

- General Comment- What is the methodology that is being used? The type of qualitative approach, relevant theory/frameworks, and research paradigm have not been addressed.

Response: We have now added the following paragraphs to the methods section (pages 6 and 8):

Interviews were considered to be the most appropriate method for this study because they provided the opportunity to encourage participants to think carefully about their own experiences (21), enable the interviewer to respond and follow up on issues raised by the interviewee (22), and that some participants may feel intimidated at the prospect of discussing their experiences within a focus group setting(23).

Thematic analysis was undertaken using the constant comparison technique of grounded theory (41, 42), which involved using the inductive identification of codes from the data to generate new hypotheses about phenomena that are derived or grounded in the data (43). Its central principle is of constant comparison, where new findings are systematically compared with existing data so that similarities and differences can be identified and emerging theories refined through the ongoing assimilation of data (41, 44).

- General Comment- The methods described are indeed commonly used in qualitative research, however, the methods section is lacking appropriate citations to demonstrate that the approach is based on established practices. The only citation in the entire methods section is for Braun and Clarke, which is a beginner's guide that is geared toward students. This section would be strengthened by the addition of appropriate references (for sampling strategies, use of semi-structured interviews, interview guide design, analytic approach etc.).

Response: Thank you for pointing this out, we completely agree that this will strengthen the methods section considerably. We have added the following information about the methods (pages 6-9):

Interviews were considered to be the most appropriate methodology for this study because they provided the opportunity to encourage participants to think carefully about their own experiences [1], enable the interviewer to respond and follow up on issues raised by the interviewee [2], and that some participants may feel intimidated at the prospect of discussing their experiences within a focus group setting[3].

Study recruitment ceased when the qualitative team agreed that theoretical saturation had been achieved (where no new themes were forthcoming from three consecutive interviews)[4, 5].

The interviewer began the discussion with more general questions to build rapport [6] such as the participants' background and general views of defining innovation. The interviewee's comments were restated and incorporated into further questions to check with the interviewer that they had understood correctly[7]. In order to gain more detailed and comprehensive understanding of the descriptions given by participants, they were asked to elaborate their comments with explanations and examples (such as, 'Why do you feel this was?')[7], or fillers (such as 'mmm' or 'yeah') were used by the interviewer to show the participant this was something they were interested in and to encourage them to continue [8]. Towards the end of the interview, participants were given an opportunity to raise issues that were important to them that had not already been covered [6].

Thematic analysis was undertaken using the constant comparison technique of grounded theory [9, 10], which involved using the inductive identification of codes from the data to generate new hypotheses about phenomena that are derived or grounded in the data [11]. Its central principle is of constant comparison, where new findings are systematically compared with existing data so that similarities and differences can be identified and emerging theories refined through the ongoing assimilation of data [9, 12]. Interview recordings were transcribed and anonymised before being imported into a data management program (NVivo version 11) to organise the data systematically, allow for quick searching and refining of codes and categories, increase the transparency of the analysis process and enable team comparisons of coding [4, 13]. First, analysts (JZ, DE & SP) read and re-read transcripts before independently coding a sample of five transcripts to begin familiarising themselves with the data [4, 14]. Interesting features were coded, whereby a segment of data was assigned a label [15], and on-going potential ideas and coding schemes were noted at every stage of analysis[16]. Codes were compared to ensure some consistency and ensure rigour in the findings[17], whereby differences were discussed thoroughly until a consensus was met and an initial coding frame was developed

We have also added the references throughout the methods section so that the following references are included:

- O'Brien, B.C., et al., *Standards for reporting qualitative research: a synthesis of recommendations*. *Acad Med*, 2014. 89(9): p. 1245-51.
- Willig, C., *Applied discourse analysis: Social and psychological interventions*. 1999: Open University Press.
- Burman, E., *Minding the Gap: Positivism, Psychology, and the Politics of Qualitative Methods*. 1997. 53(4): p. 785-801.
- Dempster, M., *A Research Guide for Health and Clinical Psychology*. 2011: Palgrave Macmillan.
- Patton, M.Q.J.Q.s.w., *Two decades of developments in qualitative inquiry: A personal, experiential perspective*. 2002. 1(3): p. 261-283.
- Fassinger, R.E.J.J.o.c.p., *Paradigms, praxis, problems, and promise: Grounded theory in counseling psychology research*. 2005. 52(2): p. 156.
- Sandelowski, M.J.R.i.n. and health, *Sample size in qualitative research*. 1995. 18(2): p. 179-183.
- Braun, V. and V. Clarke, *Successful qualitative research: A practical guide for beginners*. 2013: sage.
- Bowen, G.A., *Naturalistic inquiry and the saturation concept: a research note*. 2008. 8(1): p. 137-152.
- Rubin, H.J. and I.S. Rubin, *Qualitative interviewing: The art of hearing data*. 2011: sage.
- Hammersley, P. and M. Atkinson, *Ethnography: principles and practice*. 1983, London: Routledge.
- Smith, J.A.J.R.m.i.p., *Semi-structured interviewing and qualitative analysis*. 1995.
- Britten, N.J.B., *Qualitative research: qualitative interviews in medical research*. 1995. 311(6999): p. 251-253.
- Brinkmann, S. and S. Kvale, *Interviews: Learning the craft of qualitative research interviewing*. Vol. 3. 2015: Sage Thousand Oaks, CA.
- Fielding, N. and H. Thomas, *Qualitative interviewing*. 2008.
- Charmaz, K. and L.J.T.S.h.o.i.r.T.c.o.t.c. Belgrave, *Qualitative interviewing and grounded theory analysis*. 2012. 2: p. 347-365.
- Glaser, B.G. and A.L. Strauss, *The Discovery of Grounded Theory*. 1967, Aldine Transaction: USA.
- Seale, C. and D.J.T.E.J.o.P.H. Silverman, *Ensuring rigour in qualitative research*. 1997. 7(4): p. 379-384.
- Krippendorff, K.J.H.c.r., *Reliability in content analysis: Some common misconceptions and recommendations*. 2004. 30(3): p. 411-433.
- Miles, M.B., et al., *Qualitative data analysis: An expanded sourcebook*. 1994: sage.
- Pope, C., S. Ziebland, and N. Mays, *Analysing qualitative data*. *BMJ*, 2000. 320(7227): p.

114-116.

Braun, V. and V. Clarke, *Using thematic analysis in psychology. Qualitative Research in Psychology*, 2006. 3(2): p. 77-101.

Charmaz, K., *Constructing grounded theory: A practical guide through qualitative analysis*. 2006: sage.

Barbour, R.S.J.B., *Checklists for improving rigour in qualitative research: a case of the tail wagging the dog?* 2001. 322(7294): p. 1115-1117.

Dey, I., *Qualitative data analysis: A user friendly guide for social scientists*. 2003: Routledge.

- Design, Para 1, Sentence 3- Revise to state, “The study is reported in line with the Standards for Reporting Qualitative Research (SRQR) citation.”

Response: This has now been updated on page 8 of the methods section:

The study is reported in line with the Standards for Reporting Qualitative Research (SRQR; Additional File 1) (24).

- Sampling and Recruitment, Para 1, Sentence 5- The authors mention that individuals with experience in “first in human procedures” were included in the study. This indicates that the interviews with participants also included conversations about innovations introduced as part of clinical trials. This may have influenced study findings substantially. For example, interventions introduced as part of a clinical trials may be more readily identified by surgeons as “innovative”. Moreover, given that consent to participate in research is typically subject to unique regulatory requirements and processes (e.g., REB approval, use specific forms, etc.), it is likely that informed consent requirements in a research-context are more clearly set out compared to a non-research context. Please provide information about whether this study encompassed responses pertaining to surgical innovations introduced as research, versus those being introduced solely as clinical care (i.e., no research component), and the implications for study findings.

Response: Thank you for pointing this out. As previously mentioned, we have now added the following to the results section on page 9:

There was variation in how the procedure had been introduced in participants’ hospitals. Within this, surgeons’ own experiences included requesting approval from their trust’s new procedure committee, conducting the procedure in the context of a research study or deeming that approval was not necessary as the procedure was a variation of an existing one (from either judgement of the surgeon or confirmation from a committee). Table 2 highlights that how the challenges of defining innovation, alongside the required technical knowledge of the procedure, could cause uncertainty as to what governance was required:

“So, I’m not convinced that every innovative thing in the trust goes through the committee, because a lot of it depends on the diligence of the people who are introducing new things. So, it’s not a bad process but it’s just whether it’s comprehensive enough, I guess.” (Surgeon 1, Male, Neurology, Trust 1)

Table 2: Participants' experiences of governance for new procedures

	Example quote
Procedure considered and approved by trust’s new procedure committee	“There’s one surgeon in [European country] who described the procedure, and the surgeon in [Asian country] who had done some of these procedures, but not really clear on how they’d done it. I contacted the [European] surgeon. He gave me some pointers, so I discussed it with my colleagues, and of course put it through the governance processes as a novel and a new procedure. And then we did it.” (Surgeon 31, Colorectal, Male, Trust 8)
Approval from trust’s new procedure committee deemed unnecessary as felt to be a variation of established procedure	“We asked about it to the new procedures committee, but they said well it’s basically [variation of established procedure] isn’t it. The technicalities of it are lost on non-thoracic surgeons really, which is good.” (Surgeon 31, General surgery, Male, Trust 8) “There’s a new technique that’s probably done by no more than 9 or 10 surgeons in the country. Again, it’s a new technique, it’s only been reported in the last 3 or 4 years. [...] Because it was under the umbrella of [broader procedure], I didn’t have to go through any, so within the hospital there’s a board about new techniques, you have to sort of, clinical effectiveness group, you have to have that approved. But, because this was a variation, we didn’t need that approval.” (Surgeon 22, Male, Orthopedics, Trust 7)
Procedure evaluated in the context of a research study	“We were looking [procedure], and so that was in the guise of a randomised double-blind trial that we put through Ethics. So that would be innovative. We’ve got an Ethics Committee, and as part of that if you were going to be doing something really brand new that would have to be conducted as part of a trial, so, but when there’s a blur between doing something slightly different then no I don’t think it’s the place for that [...] I can’t think of anything where they’ve been involved in, you know, anything that we’re doing.” (Surgeon 46, Male, Orthopedics, Trust 22)

- Data Collection- The authors have identified two distinct stakeholder groups with whom interviews were conducted: (1) surgeons, and (2) governance representatives. These two groups of individuals would have very different experiences with and perspectives on informed consent in the context of surgical innovation. Please explain why each of these two stakeholder groups were selected and what information you hoped to gain from each groups

(e.g., “Governance representatives were included in this study because....”).

Response: We have now added the following information to the introduction and methods on page 6:

New invasive procedures may be introduced in the context of formal research studies or via local hospital policies (19). The National Institute for Health and Care Excellence (NICE) recommends that local NHS organisations have appropriate governance structures in place to review, approve and monitor the introduction of new invasive procedures (20).

Governance representatives (defined as those involved in regulating the introduction of new/modified procedures and/or devices at trust or national levels) were included to understand what processes were in place for the introduction of new procedures (17). Surgeons were recruited to explore experiences of introducing new/modified procedures or devices into clinical practice.

- Data Collection- The authors explain that the interview guide was developed iteratively. How did the initial interview guide change over the course of the study?

Response: We have now provided the following information about the topic guides in the section on data collection on page 7 and 8:

Separate topic guides were developed for governance representatives and surgeons. All interview schedules contained core topics of interest (e.g. ‘How would you define innovation?’, ‘What does informed consent mean to you?’)[18, 19], although these were flexible to allow participants to shape the direction of conversations[7, 20]. Governance representatives were asked how applications were processed and how these procedures are integrated into practice, whereas surgeons were about their experiences of introducing a procedure and what patients had been told. An example of an interview schedule is provided in the supplementary files (Additional files 2 and 3).

Topic guides were developed iteratively during the study period, in consideration of emerging insights from interviews and better ways of phrasing questions(29, 35). For instance, interviewers (non-clinical academics) used more prompts to encourage the surgeons to describe procedures in a non-technical manner (31). Topic guides were initially very long so were reduced to include more ‘mapping’ questions that addressed the core components of interest (such as, ‘I understand you’ve conceptualised a new procedure. Can you talk me through this?’). Additional prompts were added from emerging findings (i.e. colleagues’ reactions to procedures and written information).

- Data Collection- Related to the previous comment, the interview guide that was provided as Appendix 1 seems much more geared toward surgeons. Please provide details about how the questions/focus of the interviews differed between the two groups. How was it adapted for governance representatives?

Response: We have now included examples in the Supplementary files.

- Data Analysis- Were transcripts anonymized, or de-identified? If anonymized, please explain process of anonymization.

Response: We have now clarified that transcripts were de-identified on page 8:

Interview recordings were transcribed and de-identified so that names and identifiable information were removed.

Results

- General Comment-Consider reframing the first two themes so that they are not worded as questions.

Response: We have rephrased the theme names, as summarised on page 9:

Findings related to informed consent are presented under four main themes: difficulty defining innovation, differing views on what – and how - patients should be told, the challenges of discussing uncertainty, and managing patient expectations.

- General comment –The very first paragraph provides details on study participants, as well as information on data collection. Rather than being placed in the results section, I suggest that this information be placed in the methods section under the appropriate sub-heading (i.e., sampling and recruitment, or data collection data collection).

Response: Thank you for spotting this – we have now moved this information into the sampling and recruitment section in the methods on page 9:

Sampling and recruitment

Sampling was regularly reviewed to ensure surgeons from different specialities and geographical locations, with varying experiences of innovation (e.g. minor modifications vs first in human procedures), were recruited to capture a diversity of perspectives (30, 31). For governance representatives, maximum variation was sought in relation to different roles (e.g. new procedures committee members, medical directors), geographical locations, trust types, and different surgical specialities.

- “What is Innovation?”, fifth quote- The fifth quote under this theme seems to bring up research again (“...if you’re not trialling the instrument to see if it’s safe...”. The quote speaks to recognition that the standards for informed consent differ when the innovation is being introduced as part of clinical trial. The difference in context (research vs. non-research) needs

to be addressed by the authors in the manuscript. How do innovations introduced via research channels fit in to this study.

Response: As mentioned, we have now added a section in the results about the difference contexts that innovative procedures can be introduced (see pages 7 and 8 of this document).

- “What should patients be told?”, paragraph 2- The authors identify several types of information (evidence, experience, training, safety precautions, national guidance) as not frequently discussed by physicians, but use quotes that state that experience and evidence are in fact discussed by physicians. I would suggest selecting different quotes to better illustrate the point being made (i.e., quotes illustrating that these things are not frequently discussed).

Response: Thank you for bringing this to our attention. We have now rephrased this on page 12, and added different quotes:

Taken together, there appeared to be uncertainty at precisely how much to tell patients:

“Sometimes I tell them too much or tell them... I mean, the latest, er, ruling of Montgomery as I understand it is that they need to be told what they would want to know. Bloody hell. But then you've got to judge what they want to know and you've got to have some kind of communication with them and reasonably understand what it is they want to know.” (Surgeon 31, Colorectal, Male, Trust 8)

“There is a difficult balance I think, and I'm not sure we've got it [consent] right yet actually.” (P7)

Discussion

- Para 2, Sentence 3- Change to “innovative treatments”

Response: This has been updated on page 16 (now paragraph 3, sentence 2)– thank you for noticing this:

It is likely that the added complexity associated with innovative treatments

- Para 5- This is the first time that the authors explicitly address research. The difference in how research is regulated compared to practice changes occurring outside of the scope of research are very different. This should be addressed at the outset of the paper, and warrants substantial attention in the discussion.

Response: We have now added the following information to the introduction (page 5):

New invasive procedures may be introduced in the context of formal research studies or via local hospital policies (16). The National Institute for Health and Care Excellence (NICE) recommends that local NHS organisations have appropriate governance structures in place to review, approve and monitor the introduction of new invasive procedures (17).

On pages 11 and 12 we have also added a section to the results section if surgeons obtained approval to perform the procedure, and whether this was as part of a research study or approved by a hospital committee (see pages 7 and 8 of this document). We have also now reflected upon this in the discussion on page 15:

In addition, procedures that were deemed innovative were sometimes introduced as part of routine clinical practice without being part of an early phase study or gaining hospital approvals. Birchley et al have suggested that instead of focusing on whether something is 'innovative' or not, the potential risks and the ethical appropriateness of modifying surgical practice should be considered to ensure the safe translation of surgical innovation into clinical practice(26)

Re: Standards for reporting qualitative research

- Purpose or research question-Not explicitly stated.

Response: We have now clarified the objectives of the research on page 5:

The current study sought to explore stakeholder views of information provision for the introduction of new invasive procedures in the United Kingdom (UK).

- Qualitative approach and research paradigm-Not addressed.

Response: More information has been added on page 7 of the methods section:

Thematic analysis was undertaken using the constant comparison technique of grounded theory (38, 39), which involved using the inductive identification of codes from the data to generate new hypotheses about phenomena that are derived or grounded in the data (40). Its central principle is of constant comparison, where new findings are systematically compared with existing data so that similarities and differences can be identified and emerging theories refined through the ongoing assimilation of data (38, 41).

- Researcher characteristics and reflexivity-Not addressed.

Response: More information has been added on pages 6 and 8 of the methods section:

All interviews were conducted by one of two trained and experienced qualitative researchers (JZ or DE). JZ is a male Senior Research Associate with a background in public health research. DE is a female Research Fellow with experience working on surgical trials. She is also a member of the QuinteT research group, which uses qualitative research methods to optimise recruitment and informed consent to randomised controlled trials (RCTs). JZ and DE have several years of experience conducting qualitative research, and each has a PhD in Psychology.

Emerging findings were regularly discussed with JMB and NB (both academic surgeons), with reference to the raw data, to ensure they fully encapsulated the meaning of the data(45).

References

1. Willig, C., *Applied discourse analysis: Social and psychological interventions*. 1999: Open University Press.
2. Burman, E., *Minding the Gap: Positivism, Psychology, and the Politics of Qualitative Methods*. 1997. **53**(4): p. 785-801.
3. Dempster, M., *A Research Guide for Health and Clinical Psychology*. 2011: Palgrave Macmillan.
4. Braun, V. and V. Clarke, *Successful qualitative research: A practical guide for beginners*. 2013: sage.
5. Bowen, G.A., *Naturalistic inquiry and the saturation concept: a research note*. 2008. **8**(1): p. 137-152.
6. Brinkmann, S. and S. Kvale, *Interviews: Learning the craft of qualitative research interviewing*. Vol. 3. 2015: Sage Thousand Oaks, CA.
7. Rubin, H.J. and I.S. Rubin, *Qualitative interviewing: The art of hearing data*. 2011: sage.
8. Fielding, N. and H. Thomas, *Qualitative interviewing*. 2008.
9. Glaser, B.G. and A.L. Strauss, *The Discovery of Grounded Theory*. 1967, Aldine Transaction: USA.
10. Seale, C. and D.J.T.E.J.o.P.H. Silverman, *Ensuring rigour in qualitative research*. 1997. **7**(4): p. 379-384.
11. Krippendorff, K.J.H.c.r., *Reliability in content analysis: Some common misconceptions and recommendations*. 2004. **30**(3): p. 411-433.
12. Miles, M.B., et al., *Qualitative data analysis: An expanded sourcebook*. 1994: sage.
13. Pope, C., S. Ziebland, and N. Mays, *Analysing qualitative data*. BMJ, 2000. **320**(7227): p. 114-116.
14. Braun, V. and V. Clarke, *Using thematic analysis in psychology*. *Qualitative Research in Psychology*, 2006. **3**(2): p. 77-101.
15. Charmaz, K. and L.J.T.S.h.o.i.r.T.c.o.t.c. Belgrave, *Qualitative interviewing and grounded theory analysis*. 2012. **2**: p. 347-365.
16. Charmaz, K., *Constructing grounded theory: A practical guide through qualitative analysis*. 2006: sage.
17. Barbour, R.S.J.B., *Checklists for improving rigour in qualitative research: a case of the tail wagging the dog?* 2001. **322**(7294): p. 1115-1117.
18. Smith, J.A.J.R.m.i.p., *Semi-structured interviewing and qualitative analysis*. 1995.
19. Britten, N.J.B., *Qualitative research: qualitative interviews in medical research*. 1995. **311**(6999): p. 251-253.
20. Hammersley, P. and M. Atkinson, *Ethnography: principles and practice*. 1983, London: Routledge.

VERSION 2 – REVIEW

REVIEWER	Margaret M Byrne Moffitt Cancer Center Tampa FL, USA
REVIEW RETURNED	04-May-2020

GENERAL COMMENTS	The authors of this manuscript have done a very detailed and comprehensive revision based on the reviews that they received. They have satisfactorily addressed all of my concerns and (I believe, although up to the other reviewer) the concerns of the other reviewer as well.
---

REVIEWER	Cynthia Kendell Nova Scotia Health Authority, Canada
REVIEW RETURNED	06-May-2020

GENERAL COMMENTS	Thank you for the opportunity to review this revised manuscript. I commend the authors for their careful consideration of reviewer comments and the great deal of effort that was clearly put into revising this manuscript. It is my opinion that the manuscript has been much improved and more effectively communicates the work that was carried out. In particular, I think the revisions to the introduction and methods sections were very well done. After reading through the revised manuscript, I do have additional comments, which are listed below (most are editorial in nature). Specific comments: Title page-The title has changed. Does the running head need to change as well? Introduction-The authors have done a great job of explaining the current body of literature and identifying knowledge gaps. Introduction, paragraph 1, sentence 2- Change “a doctor’s duty to inform patients prior to surgery was used to judge on whether they had acted in line with the view of a responsible body of medical opinion” to “a doctor’s duty to inform patients prior to surgery was judged based on whether they had acted in line with the view of a responsible body of medical opinion.” Introduction, paragraph 2, sentence 4- Change “patients-clinician” to “patient-clinician”. Methods-The authors have done an excellent job of strengthening the methods section by adding more detail and incorporating relevant references. Methods, Patient and Public Involvement, sentence 2-Is it necessary to abbreviate PPI? If so, it should first be written out in full with the abbreviation in brackets. Methods, Design, sentence 2-In terms of research design, I would describe this study as a qualitative study using a grounded theory methodology. Seeing as the next heading is participants and recruitment, it’s fine to say that semi-structured interviews are being carried out, however, the justification for the use of interviews is best suited to the Data Collection section. Methods, paragraph 2, sentence 3- This may be best placed at the very beginning of the methods section, directly beneath the section heading and before any subheadings. Results, Participants, Table 1-Several points regarding this table: (1) Are the trust identification numbers meaningful outside of this study? If the numbers were assigned to the trusts by the research team please explicitly state. (2) The authors have indicated that 23
--

	trusts are represented; however, a number 24 appears in the table. (3) I'm not sure that the details of every participant are required. I suggest not including this table and focusing on creating a text description of each participant group with aggregate details, rather than individual-level. Results, Analysis, Difficulty defining innovation, paragraph 2- Replace "Table 2 highlights that how the..." with "Table 2 highlights that the." Results, Analysis, Difficulty defining innovation-The table regarding governance (which I think is helpful) is followed by a quote ("So, I'm not convinced that every innovative thing in the trust goes through the committee..."). In looking at the markup version, I believe this may have been included in error. Results, Analysis, Differing views on what-and how-patients should be told, paragraph 2-Replace "such be disclosed" with "should be disclosed". Results, Analysis, The challenges of discussing uncertainty-I realize the title of this theme has already been changed once, but I am still not sure that the current title accurately reflects the description of the theme. To be, theme seems to be about the challenges of discussion innovations with patients, not so much the challenges of discussing uncertainty (which comprises only a small portion of the description of the theme). Results, Analysis, Managing patient expectations- Regarding the sentence "Managing patients' preferences, particularly if they align with the personal preferences, appeared challenging for some", should this be "particularly if they align with the personal preferences of the surgeon"? Discussion, paragraph 2, sentence 6- Replace "This meant that patient information provided..." with "This meant that information provided to patients...". The title of Additional File 2 is "Topic guide for interviews with data committee members". Consider changing to "Topic guide for interviews with governance representatives". The title of Additional File 3 is "Topic guide for interviews stakeholders". Consider changing to "Topic guide for interviews with surgeons".
--	---

VERSION 2 – AUTHOR RESPONSE

Reviewer: 1

The authors of this manuscript have done a very detailed and comprehensive revision based on the reviews that they received. They have satisfactorily addressed all of my concerns and (I believe, although up to the other reviewer) the concerns of the other reviewer as well.

Response: Thank you for the positive feedback. We were grateful for your comments and felt they very much strengthened the manuscript.

Reviewer 2

Thank you for the opportunity to review this revised manuscript. I commend the authors for their careful consideration of reviewer comments and the great deal of effort that was clearly put into revising this manuscript. It is my opinion that the manuscript has been much improved and more effectively communicates the work that was carried out. In particular, I think the revisions to the

introduction and methods sections were very well done. After reading through the revised manuscript, I do have additional comments, which are listed below (most are editorial in nature).

Specific comments:

Title page-The title has changed. Does the running head need to change as well?

Response: Thank you for pointing this out. We have now updated the running head to 'Discussing surgical innovation with patients'. (Page 1)

Introduction-The authors have done a great job of explaining the current body of literature and identifying knowledge gaps.

Response: Thank you – your suggestions really helped.

Introduction, paragraph 1, sentence 2- Change “a doctor’s duty to inform patients prior to surgery was used to judge on whether they had acted in line with the view of a responsible body of medical opinion” to “a doctor’s duty to inform patients prior to surgery was judged based on whether they had acted in line with the view of a responsible body of medical opinion.”

Response: This has now been changed (page 5).

Introduction, paragraph 2, sentence 4- Change “patients-clinician” to “patient-clinician”.

Response: We have now changed this (page 5).

Methods-The authors have done an excellent job of strengthening the methods section by adding more detail and incorporating relevant references.

Response: Thank you.

Methods, Patient and Public Involvement, sentence 2-Is it necessary to abbreviate PPI? If so, it should first be written out in full with the abbreviation in brackets.

Response: The PPI abbreviation has been removed.

Methods, Design, sentence 2-In terms of research design, I would describe this study as a qualitative study using a grounded theory methodology. Seeing as the next heading is participants and recruitment, it’s fine to say that semi-structured interviews are being carried out, however, the justification for the use of interviews is best suited to the Data Collection section.

Response: We have added the following to the ‘design’ section so that it includes your suggestion about grounded theory methodology:

This qualitative study consisted of semi-structured interviews and used a grounded theory methodology. (Page 6)

We also moved the justification for the use of interviews to the ‘data collection’ section (page 7).

Methods, paragraph 2, sentence 3- This may be best placed at the very beginning of the methods section, directly beneath the section heading and before any subheadings.

Response: We have now moved this to the beginning of the methods section (page 6).

Results, Participants, Table 1-Several points regarding this table: (1) Are the trust identification numbers meaningful outside of this study? If the numbers were assigned to the trusts by the research team please explicitly state. (2) The authors have indicated that 23 trusts are represented; however, a number 24 appears in the table. (3) I’m not sure that the details of every participant are required. I suggest not including this table and focusing on creating a text description of each participant group with aggregate details, rather than individual-level.

Response: We had assigned the trust numbers but agree that they are not meaningful outside of this study. The '24' in the table had been an error – there are 23 trusts. We agree with your suggestion about removing the table, and in line with the editor's comments too, have now deleted this.

Information that was not included in the participant section, but was initially in the table, has now been added in bold:

*Of the 83 participants that were approached, 42 participants were recruited (those who did not participate were unable to find time for an interview or did not respond to study invites). The final sample included 26 surgeons and 16 governance representatives from across 23 NHS Trusts. Trusts varied in geographical location, type/size (i.e. teaching, specialist, large, multi-centre) and foundation status. Governance representatives' roles within trusts and NICE varied (e.g. medical director, director for quality improvement, head of governance), however, all played a role in regulating the introduction of new surgical procedures and/or devices. At the time of interview, all but one (retired) surgeon worked as a consultant within the NHS. **Surgical specialities included cardiothoracic, gastrointestinal, breast, ophthalmology, neurosurgery, urology and orthopaedics. Participants were mostly male (87%).** Interviews were conducted over the phone (55%) or face-to-face at participants' place of work or at a university. Interviews lasted an average of 43 minutes (range 22 – 112 minutes, standard deviation 17 minutes). (Page 9)*

Results, Analysis, Difficulty defining innovation, paragraph 2-Replace "Table 2 highlights that how the..." with "Table 2 highlights that the."

Response: Thank you for spotting this, we've updated this on page 10.

Results, Analysis, Difficulty defining innovation-The table regarding governance (which I think is helpful) is followed by a quote ("So, I'm not convinced that every innovative thing in the trust goes through the committee..."). In looking at the markup version, I believe this may have been included in error.

Response: We have now removed this quote (page 10).

Results, Analysis, Differing views on what-and how-patients should be told, paragraph 2-Replace "such be disclosed" with "should be disclosed".

Response: We now corrected this typo – thank you (page 11).

Results, Analysis, The challenges of discussing uncertainty-I realize the title of this theme has already been changed once, but I am still not sure that the current title accurately reflects the description of the theme. To be, theme seems to be about the challenges of discussion innovations with patients, not so much the challenges of discussing uncertainty (which comprises only a small portion of the description of the theme).

Response: We see your point. We have now renamed the theme, 'The challenges of discussing innovation' so that it is clearer and more accurately reflects the key messages. (Page 12).

Results, Analysis, Managing patient expectations- Regarding the sentence "Managing patients' preferences, particularly if they align with the personal preferences, appeared challenging for some", should this be "particularly if they align with the personal preferences of the surgeon"?

Response: Thank you for highlighting this error. We have updated the text to your suggestion (Page 15).

Discussion, paragraph 2, sentence 6- Replace "This meant that patient information provided..." with "This meant that information provided to patients..."

Response: This has been updated (page 16).

The title of Additional File 2 is “Topic guide for interviews with data committee members”. Consider changing to “Topic guide for interviews with governance representatives”.

Response: The title has now been updated.

The title of Additional File 3 is “Topic guide for interviews stakeholders”. Consider changing to “Topic guide for interviews with surgeons”.

Response: The title has now been updated.

VERSION 3 – REVIEW

REVIEWER	Cynthia Kendell Nova Scotia Health Authority, Canada
REVIEW RETURNED	15-Jun-2020

GENERAL COMMENTS	Thank you for inviting me to review these additional revisions. I have read through the manuscript, and again appreciate the effort that has gone into these changes. Several additional changes are required to the most recently uploaded version of the manuscript, which is due in part to the fact that one of my comments was not well communicated. My apologies for not being clear. I have attempted to provide clarification in my comments below. Methods  The authors stated in their response that the ethics statement could now be found under “Methods”, but it does not seem to appear anywhere in the manuscript (likely just a copy and paste error). Please ensure this statement is included. Methods, Design  My previous comment about this section was not clear. In order to be clearer, I have provided an example of how this section may be structured and worded. Please be sure to include an appropriate citation for grounded theory. “This qualitative study employed a grounded theory methodology [citation]. Grounded theory methodology was chosen because/is appropriate because_____. Consistent with grounded theory methodology, semi-structured interviews were conducted with relevant stakeholders.” Currently the following text appears under Design: Governance representatives (defined as those involved in regulating the introduction of new/modified procedures and/or devices at trust or national levels) were included to understand what processes were in place for the introduction of new procedures 17. Surgeons were recruited to explore experiences of introducing new/modified procedures or devices into clinical practice. As there is no standardised definition for innovation and it is often a continuum from standard practice through to true innovation 14 21 22, and because surgical innovation is unlikely to involve a single discrete development 23 24, participants were asked to reflect what the term meant to them and describe procedures that they deemed to be innovative. This information feels out of place under Design. I suggest moving this text back to where it was previously found, under Participants and Recruitment. Methods, Data Collection
---

	 The authors have included the following passage under Data Collection: Interviews were considered to be the most appropriate method for this study because they provided the opportunity to encourage participants to think carefully about their own experiences 36, enable the interviewer to respond and follow up on issues raised by the interviewee 37, and that some participants may feel intimidated at the prospect of discussing their experiences within a focus group setting²⁷ I have two comments about this passage of text. First, I suggest moving it to the very beginning of the Data Collection subsection. As such, the first several paragraphs under Data Collection would appear in the following order: 1) “Interviews were considered to be the most appropriate...” 2) “All participants received...” 3) “The interviewer began the discussion...” My second comment is that I suggest removing “and that some participants may feel intimidated at the prospect of discussing their experiences within a focus group setting²⁷” and adding the following sentence in its place “ Interviews were chosen instead of focus groups as it was felt that some individuals may feel intimidated at the prospect of discussing their experiences within a focus group setting.”
--	---

VERSION 3 – AUTHOR RESPONSE

Reviewer’s comments: Thank you for inviting me to review these additional revisions. I have read through the manuscript, and again appreciate the effort that has gone into these changes. Several additional changes are required to the most recently uploaded version of the manuscript, which is due in part to the fact that one of my comments was not well communicated. My apologies for not being clear. I have attempted to provide clarification in my comments below.

***Response:** We are very pleased to see that the changes to the paper were well-received and we appreciate the reviewer’s helpful and constructive comments. We have listed our response to each point raised by the reviewer and highlight the changes made to the manuscript. We look forward to your response.*

Methods

The authors stated in their response that the ethics statement could now be found under “Methods”, but it does not seem to appear anywhere in the manuscript (likely just a copy and paste error). Please ensure this statement is included.

***Response:** This was included on page 6, in the Design section under ‘Methods’:*

‘Ethical approval was granted by the University of Bristol Faculty of Health Sciences Research Ethics Committee.’ (Page 6)

Methods, Design

My previous comment about this section was not clear. In order to be clearer, I have provided an example of how this section may be structured and worded. Please be sure to include an appropriate citation for grounded theory.

“This qualitative study employed a grounded theory methodology [citation]. Grounded theory methodology was chosen because/is appropriate because_____. Consistent with grounded theory methodology, semi-structured interviews were conducted with relevant stakeholders.”

Response: We have now updated the section in line with the reviewer's suggestion and included an appropriate citation.

This qualitative study employed a grounded theory methodology (21) because it enabled the inductive identification of codes from the data to generate new hypotheses about phenomena that are derived or grounded in the data (22). Its central principle is of constant comparison, where new findings are systematically compared with existing data so that similarities and differences can be identified and emerging theories refined through the ongoing assimilation of data (21, 23). Consistent with grounded theory methodology, semi-structured interviews were conducted with relevant stakeholders. (Page 6)

Currently the following text appears under Design:

Governance representatives (defined as those involved in regulating the introduction of new/modified procedures and/or devices at trust or national levels) were included to understand what processes were in place for the introduction of new procedures 17. Surgeons were recruited to explore experiences of introducing new/modified procedures or devices into clinical practice. As there is no standardised definition for innovation and it is often a continuum from standard practice through to true innovation 14 21 22, and because surgical innovation is unlikely to involve a single discrete development 23 24, participants were asked to reflect what the term meant to them and describe procedures that they deemed to be innovative.

This information feels out of place under Design. I suggest moving this text back to where it was previously found, under Participants and Recruitment.

Response: This has now been done, so that the text is under 'Sampling and recruitment' (see last paragraph on Page 6).

Methods, Data Collection

The authors have included the following passage under Data Collection:

Interviews were considered to be the most appropriate method for this study because they provided the opportunity to encourage participants to think carefully about their own experiences 36, enable the interviewer to respond and follow up on issues raised by the interviewee 37, and that some participants may feel intimidated at the prospect of discussing their experiences within a focus group setting²⁷

I have two comments about this passage of text. First, I suggest moving it to the very beginning of the Data Collection subsection. As such, the first several paragraphs under Data Collection would appear in the following order:

- 1) "Interviews were considered to be the most appropriate..."
- 2) "All participants received..."
- 3) "The interviewer began the discussion..."

Response: Thank you, we agree this is much clearer. We have now updated the paragraphs so that they are in the suggested order (see Page 7).

My second comment is that I suggest removing "and that some participants may feel intimidated at the prospect of discussing their experiences within a focus group setting²⁷" and adding the following sentence in its place "Interviews were chosen instead of focus groups as it was felt that some individuals may feel intimidated at the prospect of discussing their experiences within a focus group setting."

Response: We have now done this, so that the text reads:

Interviews were considered to be the most appropriate method for this study because they provided the opportunity to encourage participants to think carefully about their own

experiences (35), enable the interviewer to respond and follow up on issues raised by the interviewee (36). Interviews were chosen instead of focus groups as it was felt that some individuals may feel intimidated at the prospect of discussing their experiences within a focus group setting (26). (Page 7)

VERSION 4 – REVIEW

REVIEWER	Cynthia Kendell Nova Scotia Health Authority, Canada
REVIEW RETURNED	03-Sep-2020
GENERAL COMMENTS	Thank you to the authors for your careful consideration of the feedback provided throughout the review process. All of my comments have been satisfactorily addressed in the most recent version of the manuscript. I have no additional comments.